

# Fram Strait sea ice export variability and September Arctic sea ice extent over the last 80 years

Lars H. Smedsrud [1,2,3], Mari H. Halvorsen [1,4], Julienne C. Stroeve [5,6], Rong Zhang [7], and Kjell Kloster [8]

[1] Geophysical Institute, University of Bergen, Bergen, Norway
[2] Bjerknes Centre for Climate Research, Bergen, Norway
[3] University Centre in Svalbard, Longyearbyen, Svalbard
[4] Norwegian Mapping Authority Hydrographic Service, Stavanger, Norway
[5] National Snow and Ice Data Centre, University of Colorado, USA
[6] Centre for Polar Observation and Modelling, University College London, London, UK
[7] Geophysical Fluid Dynamics Laboratory, National Oceanic and Atmospheric Administration, Princeton, New Jersey, USA
[8] Nansen Environmental and Remote Sensing Center, Bergen, Norway

*Correspondence to*: Lars H. Smedsrud (Lars.Smedsrud@uib.no)

**Abstract.** The Arctic Basin exports between 600,000 and 1 million km² of it's sea ice cover southwards through Fram Strait each year, or about 10 % of the sea-ice covered area inside the basin. During winter, ice export results in growth of new and relatively thin ice inside the basin, while during summer or spring, export contributes directly to open water further north that enhances the ice-albedo feedback during summer. A new updated time series from 1935 to 2014 of Fram Strait sea ice area export shows that the long-term annual mean export is about 880,000 km², with large inter-annual and multidecadal variability, and no long-term trend over the past 80 years. Nevertheless, the last decade has witnessed increased ice export, with several years having annual ice export that exceed 1 million km². Evaluating the trend onwards from 1979, when satellite based sea ice coverage became more readily available, reveals an increase in annual export of about +6 % per decade. The observed increase is caused by higher southward ice drift speeds due to stronger southward geostrophic winds, largely explained by increasing surface pressure over Greenland. Spring and summer area export increased more (+11 % per decade) than in autumn and winter (+2.6 % per decade). Contrary to the last decade, the 1950 – 1970 period had relatively low export during spring and summer, and consistently mid-September sea ice extent was higher during these decades than both before and afterwards. We thus find that export anomalies during spring have a clear influence on the following September sea ice extent in general, and that for the recent decade, the export may be partially responsible for the accelerating decline in Arctic sea ice extent.





## 1 Introduction

Along with expectations for a warming planet, the spatial extent of the Arctic sea ice cover has declined. This is especially apparent during the last two decades as the Arctic sea ice cover has become both thinner and smaller in extent (Comiso 2012, Stroeve et al. 2012). In September 2012, the lowest September sea ice extent (SIE) since the satellite record started in 1979

occurred. The 2012 minimum was 16 % lower than in 2007, and 44 % below the 1981-2010 average minimum. While the ice recovered somewhat after 2012, the 9 lowest September sea ice extents have all occurred in the last 9 years. A number of processes have been suggested to explain parts of the sea ice loss, but both observations and simulations from global climate model point to the increased greenhouse gas forcing as the main driver of the observed sea ice loss (Stroeve and Notz, 2015; Kay et al., 2011). Natural variability has also played a role, including increased poleward transport of heat in both the ocean

and atmosphere (Graversen et al 2011, Zhang 2015), and an increase in longwave radiation due to cloud cover (Francis et al., 2005). Yet, despite the large number of existing studies, the role and influence of natural variability remains unclear, especially on the longer time-scales expanding further back than the last 30 years.

Changes in ice export are considered an important contributor to SIE variability (Nghiem et al., 2007; Smedsrud et al., 2011)

and perhaps also in regards to the observed long-term trends. Historically, about 10 % of the Arctic sea ice area is exported through Fram Strait (FS) annually (Fig. 1), and the ice export through the other Arctic gateways are an order of magnitude smaller (Kwok 2009). Because quite thick ice is lost by this export through FS (Hansen et al., 2013), a larger than normal ice export will decrease the remaining mean thickness within the Arctic Basin. An influence of export anomalies on Arctic sea ice thickness was previously suggested by Rigor et al. (2002) using buoy data. A similar conclusion was reached using

model simulations from climate models participating in the Coupled Model Intercomparison Project Phase 5 (CMIP5) (Langehaug et al., 2013). Recently Fučkar et al. (2015) found that much of the Northern Hemisphere sea ice thickness variability could be explained by changes in sea ice motion related to wind forcing.

Several studies have suggested that sea ice drift speeds are increasing, both within the Arctic Basin (Hakkinen et al., 2008;

Rampal et al., 2009), and also in FS (Rampal et al., 2009, Smedsrud et al., 2011). Positive trends were also found in the annual FS ice area export by Widell et al. (2003) (4 % per decade from 1950 - 2000) and Smedsrud et al. (2011) (5 % per decade from 1957 - 2010). Using the available NSIDC sea ice drift data, Krumpen et al (2016) recently found a much higher trend for 1980-2012 (37.6 % per decade), but noted that the large positive trend seemed "unrealistic". However, contrary to these studies, Kwok et al., (2013) found a small negative trend in annual FS ice area export between 1982 and 2009, but with

positive trends for 2001 – 2009 for both annual (October - September) and summer (June - September). Spreen et al. (2009) did not observe any significant change in FS ice volume export for the period  2003 – 2008 for observed winter means (October – April). Thus, there remains some uncertainty exactly how much export through FShas changed and how it has influenced the long-term decline in the summer ice cover.



The Arctic seasonal maximum sea ice cover generally occurs in late February or early March (Zwally and Gloersen 2008), though it has also been observed to occur as late as early April (e.g. on April 2 in 2010). Changes in ice export through FS between March and August could therefore influence the following September SIE by fostering development of open water within the icepack that in turn enhances the ice-albedo feedback during the melt season (Smedsrud et al., 2011; Kwok and Cunningham 2010). Such an influence has recently been examined between 1993 and 2012 by Williams et al. (2016) in combination with coastal divergence. This work expands on that by Williams et al. (2016) by estimating the FS ice area export from 1935 to 2014. We evaluate the long-term mean, variability and trends over this 80 year record, and further examine the influence of the long-term FS export on a new time-series of September SIE, also covering the years 1935-2014 (Walsh et al. 2015).

## 2 Data and Methods

### 2.1 Ice Drift observations: 2004-2014

In this study we use observed sea ice drift speeds onwards from February 2004 and updated through December 2014. The drift was calculated by recognizing displacement vectors manually on Envisat Advanced Synthetic Aperture Radar (ASAR) WideSwath and Radarsat-2 ScanSAR (from 2012) images captured 3 days apart (Kloster and Sandven 2014). Displacement vectors that cross 79°N were linearly interpolated to bins (1° longitude, each 21 km) from 15°W to 5°E (Fig. 3).

For most 3 days image pairs, displacement vectors with an accuracy of about ±3 km were found with a spacing of 30 − 50 km. This accuracy of about ±3 km per vector is considered sufficient for most analysis. This is because subsequent averaging or addition in time/space of many unbiased vectors will generally result in improved accuracy. We only used monthly mean cross-strait ice drift speed values, defined as the spatial-temporal mean southward speed of all ice crossing 79°N (Fig. 1) between the fast ice edge and the pack ice edge at 50 % sea ice concentration. On the western side of the strait, a linear interpolation from zero motion in the stable fast ice to the first measured motion vector was made. It was assumed that ice displacement to the east of the last measured vector is constant near the ice edge. The monthly mean speed value results from the averaging of about 50 individual, unbiased displacement vectors, thus the calculated mean speed value should be considerably more accurate than 10 %.

Using the mean drift speeds as derived above, corresponding FS ice area export along 79°N, is calculated as the product of sea ice drift and passive microwave sea ice concentration (Kloster and Sandven 2014). The values we use in this study are cumulative values, i.e. summarized over the month, season or year. This means that a spring ice area export value of 500,000 km² is the sum of 60 3-day values from March 1 to August 31. Alternatively mean export values could be used, i.e. this number divided by the number of days, or months. From here on, ice area export will be referred to as ice export.



**2.2 Sea Level Pressure observations: 1935-2014**

To extend the time-period, observed monthly mean Sea Level Pressure (mSLP) values were used onwards from 1935 to estimate ice export prior to the ASAR data starting in February 2004. The cross-strait difference along 78°N was calculated

between 18°W and 15°E based on monthly mSLP observations from Longyearbyen (Fig. 3, Svalbard Airport, Norwegian Meteorological Institute, http://eklima.met.no) and from weighted averages of monthly mSLP from two nearby stations on the Greenland side at 18°W; Danmarkshavn and Nord (Fig. 3, Danish Meteorological Institute, Cappelen 2014). mSLP is available from Danmarkshavn and Nord back to 1958. For the 1935-1958 period, a linear regression between Nord and Tasiilaq further south was performed.

The mSLP observations were then used to calculate cross-strait geostrophic winds following (Thorndike and Colony 1982). Because mSLP from Danmarkshavn and Nord correlated well (r=0.93), we derived a linearly interpolated value at 78°N,18°W directly using these stations onwards from 1958. For the period 1935-1958, interpolated values between station Nord and Tasiilaq were used, which have a somewhat lower correlation (r=0.77). Our method assumes that wind and ocean

drag are the dominant forces acting on the sea ice, consistent with geostrophic winds explaining more than 70 % of the variance of ice drift speed in the Arctic Ocean (Thorndike and Colony 1982).  In the FS, winds have also been found to be the dominant force acting on sea ice (Widell et al., 2003), and the cross strait pressure difference in SLP well represents the ice drift on a daily time-scale (Tsukernik et al. 2009).

**2.3 Blended Ice Drift and Export: 1935-2014**

Evaluation of the relationship between the observed SAR ice drift speed and the geostrophic wind since 2004 supports the use of mSLP to extend ice export back in time. A linear regression between monthly mean ice drift speed and geostrophic wind (r=0.77, with 95 % confidence interval [0.68, 0.83]) reveals that the ice in FS generally drifts at a speed that is 1.6 % of

the geostrophic wind speed (Eq. 1). The constant contribution resulting from the linear regression represents the speed of the ice given no local wind forcing, and is 6.7 cm/s (eq.1). In other words, the value of 6.7 cm/s represents the mean ocean current, though nonlinear components of ice drift, including forces from variations in ocean currents or internal ice stress may also represent parts of this constant (Thorndike and Colony 1982). It is important to note however, it is not the locally wind-driven, ocean current.  Both the mean drift speed and the mean ocean current are comparable to previous studies

(Widell et al., 2003; Smedsrud et al. 2011; Thorndike and Colony 1982). The standard error of the regression is 3.4 cm/s.

$$V_{ice} = 0.016 \times V_g + 0.067 \ [m/s]. \hspace{3cm} (1)$$

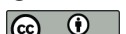



If sea ice concentration does not change systematically inside the ice pack locally, we expect a similar relationship between mSLP and sea ice export. This was indeed what we find, with a correlation between the cross-strait mSLP and ice export of r=0.73.

The annual cycle of mSLP-based ice speed and export is similar to earlier estimates, with higher speeds during winter, and weaker during summer (Kwok 2009). The annual mean speed is close to 12 cm/s (Fig. 2), which is a spatially averaged value between 15°W and 5°W, and a temporal average for the years 2004-2014.

However, a clear seasonal difference is observed between the SAR ice speed and the mSLP-estimated ice speed (Fig. 2). Previously it was assumed that the mean ocean current would be constant throughout the year (Kwok 2009; Smedsrud et al. 2011), e.g. 6.7 cm/s from eq. 1. However, based on the 10 years of detailed SAR velocities, we instead find it is necessary to account for the seasonality of the mean ocean current in order to match the mSLP-derived ice drift with the SAR data. The suggested seasonal change is a mean winter current (December – April) of 9.5 cm/s, and a mean summer current of 3.9 cm/s

(June – October). The East Greenland Current (EGC) thus appears 2.8 cm/s stronger than the mean during winter, and 2.8 cm/s weaker during summer. Note that this seasonal difference cannot be explained by a seasonally varying internal ice stress, because ice is thicker and denser during winter, which would result in a larger ice stress and therefore weaker ice drift speed for a similar wind speed.

An increase in the EGC would be consistent with generally stronger winds in the North Atlantic region during winter. This suggests that the EGC is responding to the larger scale wind-forcing as well as to the local winds. Generally, the entire circulation along the continental slope of the Arctic Basin - Nordic Seas is driven by the wind-stress curl north of the Greenland-Scotland ridge (Isachsen et al., 2003). Two recent studies confirm that the EGC is stronger during winter, and is responding to the large-scale wind stress curl in the Nordic Seas. It is thus likely that this increase is causing the additional

winter export (Fig. 2). De Steur et al. (2014) analyzed mooring data along 79°N between 1997 and 2009, and found that surface currents were below 5 cm/s during summer and 10-15 cm/s during winter, also varying in the east-west direction. Daniault et al. (2011) found a maximum in the flow in January and a minimum in July for the years 1992-2009 based on satellite radar altimetry data at 60°N, and that the vertical distribution remained constant over this time period.

The above studies support a bias-correction for the constant EGC speed in eq. 1 to increase (decrease) the mSLP-based winter (summer) ice speeds. Thus, assuming a stronger EGC during winter, and weaker during summer, we added the seasonal difference to the time-series of mSLP-based ice speed. This means that in eq. 1, we add 2.8 cm/s to the constant 6.7 cm/s for the months December through April, and subtract 2.8 cm/s from the constant 6.7 cm/s for June through October, while May and November remained unchanged. This bias corrected mSLP-based ice speed better matches the SAR



observations (Fig. 2), with a correlation of r=0.88. The same correction was also applied for the calculated ice export, representing a decrease in summer values of 23,800 km² and increase in winter values of 22,400 km² accordingly (not shown). The seasonal correction further improves the correlations between observed and mSLP-based ice export (r=0.87). In other words, we expect our bias-corrected mSLP based time-series from 1935 to 2004 to explain about 80 % of the "true" ice drift and export variability.

Using the seasonally varying EGC explained above, in addition to the monthly varying geostrophic winds based on observed mSLP, we calculate the monthly mean ice export prior to 2004 and blend them with the SAR-based observed ice export from 2004 to 2014. This generates an 80-year long record of monthly mean FS ice export.

## 2.4 Sea Ice Extent

Finally, a newly blended historical and modern record of sea ice concentrations is now available from the National Snow and Ice Data Center (NSIDC), the "Gridded monthly sea ice extent and concentration, 1850 onwards" (Walsh et al., 2015). This data set is an improvement upon an earlier historical record from Chapman and Walsh (1991), and provides mid-month sea ice concentrations on a 0.25 x 0.25 degrees grid. A total of 16 different sources of information were used to construct ice cover information back to 1850. Prior to the modern satellite data record, which began in October 1978 from a series of successive passive microwave sensors (e.g. the Scanning Multichannel Microwave Radiometer (SMMR) and several Special Sensor Microwave/Imager (SSM/I) and SSMIS sensors), observations come from earlier satellite missions, aircraft and ship observations, compilations by naval oceanographers, ice charts from national ice services, and whaling log records, among others. For many regions and time-periods several sources of sea ice data and weighting was applied (see http://nsidc.org/data/g10010 for details). The monthly files are intended to represent ice on the 15th or 16th of each month using the NASA Team sea ice algorithm. Using this data set, the ice extent is defined as the area covered by ice of greater than 15 % ice concentration.

Initial evaluation of the data set indicated a problem with inconsistencies in the land mask applied throughout the entire time-period. This was fixed and led to a slight reduction in the overall sea ice extent prior to the satellite data record. To evaluate how ice export influences changes in sea ice cover within the Arctic basin, we use an Arctic Ocean Domain mask as defined in Serreze et al. (2007) and compute sea ice extent within this domain only (Fig. 1). For the September SIE time-series this mainly excluded the Greenland Sea downstream of the FS ice export, where we expect high export to contribute to a larger ice cover.



## 3 Results

### 3.1 Annual Mean Ice Export Variability and Trends: 1935-2014

Figure 3 shows that the temporal mean ice drift speed is quite constant spatially across the FS eastward of 5°W, and that the speed decreases westward towards the Greenland coast. Velocities are clearly strongest during winter with mean speeds

above 20 cm/s, decreasing to less than 10 cm/s during summer eastward of 5°W. FS ice drift is in the south-southwesterly direction steered by the Greenland Coast. The ice export occurs mostly between 5°W and the Greenwich Meridian. The export is limited on the western side by the decreasing ice speed, reaching zero at 16°W, where stationary land fast ice is usually found. On the eastern side the ice export is limited by zero concentration varying from 5°W to 5°E  (not shown).

The 1935-2014 long-term annual FS sea ice export, defined as September 1 to August 31, is on average 883,000 km², and ranges from 0.6 to 1.2 million km² (Fig. 4). Winter export is defined as September through February and has a long-term mean of 528,000 km². We define spring export as March through August, and the mean value is 354,000 km². Note that the 1935-2014 long-term annual mean ice export of 883,000 km²  is 25 % higher than previously found by Kwok (2009) using data from 1979 to 2007, but similar to values from Colony and Thorndike (1984), Widell et al. (2003) for 1950-2000 and

Smedsrud et al. (2011) for the period 1957 to 2011.

The consequences of the export variability are discussed later, here we just note that since 2006, the ice export has remained higher than the long-term mean (Fig. 4), and that for 2010-2013, the annual export exceeded 1 million km². In addition, there are a number of notable export events in the 80-year time series. Note that there are no mSLP values observed during WWII,

so the seemingly constant values from 1940 to 1945 are set identical to the long term mean annual values. The lowest annual export occurred in 1946 with only 599,000 km² exported, and the highest export with 1,206,000 km² was in 1995.

The annual and seasonal trends appear robust, and no systematic difference appears by merging the mSLP based values prior to 2004 with the SAR-based values since 2004. This was first checked by comparing trends for the merged time series

(1935-2014) with those based on the observed mSLP only (1935-2014). For example, the mSLP-based annual trend is 0.2 versus 0.3 % per decade for the merged values, while for winter the values are 1.1 versus 1.6 % per decade, and spring -0.8 versus -1.3 % per decade. These long-term trends are small and not significantly different from zero (p =0.2), and are therefore not included in Fig. 4.

We also searched for specific cycles, or frequencies, in the new 80-year time series. Apart from the obvious annual cycle (Fig. 2) we could not find any special peaks in calculated spectrums of the annual, winter or spring export (not shown). The smoothed time series appeared similar for cut-off frequencies representing cycles above 10 years, so we chose to show a 20-year cut-off (or frequency of 0.05 cycles/year) in Fig. 4. Overall the variations are similar for the annual and spring export




values, while there is less long-term variability in the winter export. For the smoothed series there is a distinct peak in annual and spring export between 1951 and 1954. After 1954, there is a decrease in annual and spring export until the mid 1980's, and an overall increase onwards to 2014. Thus, there is a hint of a long-term multidecadal oscillation with a period around 70-years.

## 3.2 Ice Export Trends: 1979-2014

The increasing exports onwards from the 1980's create statistically significant positive trends for both the annual and seasonal values. For example, from 1979 to 2014 we find a positive trend in annual export of +5.9 % (p=0.025) per decade (Fig. 4). This trend is consistent with a general increase in ice drift speed observed well inside the deep Arctic Basin (Fig. 1, Spreen et al. 2011, Rampal et al., 2009), but the small number of buoys exiting in FS have precluded estimating trends there. The positive trend in annual ice export from 1979 to 2014 is largely driven by higher ice export during spring: the winter ice export trend is +3.0 % (p=0.213) per decade, while the spring export trend is +11.1 % (p=0.011) per decade (Fig. 4).

The increasing spring export since 1979 may have important consequences, and we therefore analyze this time-period further to make sure no biases are introduced by using the blended mSLP + SAR export fields. Trends are found to be similar between the merged (mSLP + SAR) and mSLP based time-series, with a positive trend in spring export of 13.1 % (±6.8) per decade from the mSLP only based estimates, and a trend of 11.1 % (±8.1) per decade using the merged time-series. These trends are not significantly different because the 95 % confidence intervals overlap (given in parenthesis). Note that around 1980, the spring export was approximately half of the winter export. The robust trend in spring export since 1980 has resulted in a smaller seasonal difference, and since 2011, the export in winter and spring are of similar magnitude (Fig. 4). The recent high values could perhaps be surprising, were it not for the longer time-series where a similar high spring export is evident in the 1950's.

We also do not find a "shift" in the trends after 2004 when the SAR values are used. For example, the spring 1979 - 2003 trend is +9.1 % (±11.3) per decade, almost as high as the 11.1 % for the 1979-2014 period. So the spring export is the main cause of increased annual export before and after 2003, and the differences are not significant at the 95 % confidence level. Due to the few years and large variability, the 1979 – 2003 trends are also not significantly different from zero (p>0.05). The annual trend for 1979 – 2003 is 3.8 % (p=0.36), compared to 5.9 % for 1979-2014, and for winter 0.8 % (p=0.84), compared to 2.6 % for 1979 – 2014. This indicates that the export trends since 1979 are related to a gradual increase in mSLP across FS over most of this period. The increased spring ice export is due to stronger geostrophic winds, driven by an increase of 0.53 hPa per decade in mSLP over Greenland between 1979 and 2014. The increase in SLP is strongest in June – August, and covers the larger part of Greenland (not shown). The mSLP trend on the Svalbard side is a slightly lower and negative trend, is strongest in March – May, and covers the larger part of the Barents Sea (not shown).



## 4 Discussion

In this study we use station mSLP data, rather than values derived from atmospheric reanalysis data sets (Smedsrud et al., 2011, Widell et al., 2003; Kwok 2009), because we discovered unexplained systematic differences between NCEP reanalysis mSLP fields and observed mSLP within the FS in recent years. Despite the wide use of reanalysis data sets, they are heavily influenced by the numerical model used to simulate the fields, and they are therefore regarded as less accurate than the station data used in this study.

Prior to 2004 we do not utilize observations of cross-strait variations in the width of the ice covered area, ice speed or ice concentrations, but base our ice export values solely on the regression equation found between observed mSLP from Longyearbyen, station Nord, Danmarkshavn and Tasiilaq (Fig. 3) and observed SAR ice export. While some of these values could be estimated for parts of the 1935 - 2014 period, there are in general no observations other than sea ice extent available so far back in time. Using the observed mSLP values is therefore the only consistent way to calculate ice export over the last 80-years.

### 4.1 Long-term variability of September SIE

The mid-September SIE time series shows two stages, a modest increase from 1935 until around 1965, and then a monotonic decrease over the last 50 years (Fig. 5). There is a "break-point" around the mid 1990's when the September SIE loss accelerates as has been noted earlier (Stroeve et al., 2012). From the mid 1960's until the mid 1990's the loss in SIE is small. The minimum SIE value pre-dating 1995 occurs in 1952, and the last two minima in 2007 and 2012 are also clearly visible (Fig. 5). The overall mid-September SIE maximum occurred in 1963.

Evaluating 30-year trends for successive 30-year periods, along with the 2□ standard errors shows distinct periods of ice loss and ice gain (Fig. 6). From the 1950s to 1967, trends are positive and then become negative. The long-term 1935-1990 trend is essentially 0, in stark contrast to the 1991 to 2013 trend, which is strongly negative. The last 7 consecutive 30-year trends are significantly different from trend periods before 1990 (Fig. 6).

The smoothed annual and spring export starts decreasing onwards from 1952 (Fig. 4), so this may have contributed to the increasing mid-September SIE until 1965. Since the mid 1980's the smoothed FS export has increased, consistent with the observed decease in SIE. Over the last 20 years the export has primarily increased during spring, consistent with the accelerating loss of mid-September SIE (Fig. 4 and Fig. 5).





## 4.2 Effects of long-term variability and trends in ice export

Our ice export values are largely consistent with previous studies on FS ice export for the recent decades (Kwok et al. 2013; Spreen et al. 2009; Smedsrud et al. 2011). The year-to-year variability is of the same order, and the maximum and minimum values are also similar. The largest difference to the Smedsrud et al. (2011) export values is that the time series is updated to

2014 and now extends back to 1935. In addition, there is no overall long-term linear trend in annual export.

The effect of sea ice drift variability on the Arctic sea ice cover in general has been recognized for a long time (Thorndike and Colony 1982). Rigor et al (2002) used drifting buoy data from 1979 to 1998, and found a systematic change between the 1980's and 1990's driven by the large scale atmospheric forcing. During the 1980's the Beaufort gyre was large, the ice

stayed inside the Arctic Basin for several years, and FS sea ice export was low, contributing to a thicker ice cover. In the 1990's the Beaufort gyre weakened, ice drift was more directly from the Siberian coast to FS, and the FS sea ice export was higher. Our results are consistent with Rigor et al. (2002) in that the annual export was lower during the 1980's (810,000 km²) than during the 1990's (890,000 km²). The overall maximum annual export in a calendar year occurred in 2012 with a value of 1,176,000 km², but the second largest calendar year export occurred in 1995 (1,131,000 km²). Note that these values

are a little different from those plotted in Figure 2, which show the winter + spring export from September 1 through August 31.

One suggested mechanism for the rapid decline in summer Arctic SIE is that a larger winter export could create a larger fraction of thin first year ice that is more prone to melting out the following summer. In addition, first year ice is smoother

than thick and old ice, and may allow for larger fractions of melt ponds during summer (Landy et al., 2015). Schröder et al. (2014) found a strong correlation between such simulated spring melt pond fraction and September Arctic SIE. However, in this study we find that the correlation between winter ice export and the following September SIE is modest ($r=-0.26$ between 1979-2014.). Thus, the small increase in winter ice export over the last 35 years (2.6 % per decade) suggests that summer ice loss is not particularly sensitive to winter sea ice export. Because the winter export is larger than the spring

export (Fig. 2) there has generally been a clear connection between annual and winter export anomalies. Yet while there is little change in winter export, there has been a notable increase in the spring export, i.e. after the Arctic seasonal maximum SIE occurs in late February or early March. In fact in recent years, the spring export has been almost as large as the winter export (Fig. 4). We turn our attention towards the increasing spring export in section 4.4, but first examine the cause of the variability in the larger atmospheric circulation.

## 4.3 FS ice export and the large-scale atmospheric circulation

Rigor et al. (2002) concluded that annual FS export correlated well with the Arctic Oscillation (AO) index during the 1980's and 1990's, and found a 10 % increase of FS export with an AO index of +1. The response was most apparent for the winter





(DJF) AO index and the winter ice export. Examining our longer time series, this relationship does not appear stationary in time, and since 2000, the AO index has fluctuated around zero, while the FS export has remained at anomalously high levels. The maximum DJF AO value occurred in 1989, not related to a peak in the annual FS export. Over the 80 year time-series we find that the winter AO index is not a good indicator of winter FS ice export, the correlation is as low as *r=0.19* (using

DJF AO, and winter export (SONDJF), and only increases to *r=0.22* if the winter AO index is also calculated for SONDJF. This is not surprising because the updated AO spatial pattern does not exhibit strong pressure gradients in the FS (not shown).

The dependency of FS export to a dipole pattern in the SLP and not on the AO was confirmed for the period 1979–2006 by

Tsukernik et al. (2009), studying daily FS ice export values compared to atmospheric SLP forcing from re-analysis. They found that on a daily time scale, the atmospheric circulation pattern responsible for the high export is a dipole between the Barents Sea (low pressure) and Greenland (high pressure). The ice motion was maximized at 0-lag, persisted year-round, and over time scales of 10–60 days. This SLP dipole pattern emerged from the second empirical orthogonal function (EOF) of daily SLP anomalies in both winter and summer, with maximum correlation east and west of the FS. An implication of this

result is to use station based observed cross-strait SLP pressure gradient like we have done here.

The observed cross-strait SLP gradient, the dipole pattern found by Tsukernik et al. (2009) and the Arctic Dipole (AD) are similar expressions of varying strength of the southerly winds in FS. The AD has been suggested previously as a major driver of record low Arctic summer SIE (Wang et al. 2009). The AD index is defined as the second leading mode (PC2) of

spring (April-July) SLP anomalies within the Arctic Circle. Here a positive AD is defined as having a positive SLP anomaly over Greenland and a negative SLP anomaly over the Kara and Laptev Seas, which is efficient in causing enhanced transpolar ice drift. In future projections, high rates of summer Arctic sea ice loss are also associated with enhanced transpolar drift and FS ice export driven by changing sea level pressure patterns (Wettstein et al. 2014). Wettstein at al. (2014) found co-varying atmospheric circulation patterns resembling the AD, with maximum amplitude between April and

July.

We calculated the AD index onwards from 1948 using the NCEP/NCAR reanalysis data, for which data are not available before 1948. The observed AD index and spring export correlates ($r_{AD-FS\ ice\ export} = 0.45$) over the longer period 1948-2014 (Fig. 7). The co-variability is similar over time ($r_{AD-FS\ ice\ export} = 0.44$ for 1979-2014). The AD can therefore explain

parts of the FS ice export variability, but the cross-strait SLP gradient remains the best predictor of the local wind forcing and therefore FS sea ice drift.



### 4.4 High annual and spring export during the last decade

Our updated time series shows large annual values of Arctic sea ice export during the last decade. The same SAR-based export values used here were previously used by Smedsrud et al. (2011) for 2004 – 2010, but 2011, 2012 and 2013 had un-reported annual exports above 1 million km² (Fig. 4). A comparison between SAR-based and passive microwave-based drift

speeds gave mostly similar values for both methods since 2007, but indicated some high export events that are only detected using the SAR-based drift speeds (Smedsrud et al., 2011). This is likely the major explanation for the difference between previous passive microwave-based export values (Kwok 2009) and our results. We believe that the cause of the differences primarily results from the coarse resolution of the passive satellite observations missing some high-speed export events during winter. We speculate that high sea-ice concentrations in the FS make it difficult to track individual sea ice floes using

the coarser resolution passive microwave images.

The Arctic Basin covers an area of about 7.8 million km², and has been fully ice covered from November through May since 1979. The annual ice export during the 1980's (~800,000 km²) was 10 % of this winter ice covered area. However, during the last seven years (2007-2014) the mean annual ice export increased to nearly 1 million km², representing 13 % of this area.

This is the relative ice export, or the large-scale divergence of the Arctic Basin sea ice cover; the export divided by the area covered by sea ice.

If the sea ice cover decreases and the export remains constant, the divergence, or the relative export, increases too. The observed increase in export represents a 30 % increase in the relative area export, but with a smaller annual mean ice covered

area inside the Arctic Basin the increase rises to about 40 %. This value is based on using a 1 million km² in export and a mean annual ice covered area inside the Arctic Basin of 7.0 million km² for the last 7 years (2007 – 2014). Such an increase in export is expected to contribute towards both a thinner and smaller Arctic ice cover in general (Langehaug et al., 2013), because older and thicker sea ice than the Arctic Basin average is transported southward through FS. During winter, the open water anomalies created within the basin quickly refreeze, and thus, an impact of the modestly increased winter ice export

since 1979 has likely been towards a thinner ice cover (Lindsay and Schweiger 2015). This is consistent with Fučkar et al. (2015) who found that a reduction of the FS winter export related to their Canadian-Siberian Dipole cluster explains a thickening over most of the Arctic Basin between 0.2 – 0.5 m. This cluster is the one with the largest change in FS export and explains 28.6 % of the variability, while the other two clusters mostly describe divergence within the Basin.   However, the largest FS export increase since 1979 has been during spring.

### 30 4.5 Quantification of spring ice export anomalies

The later in the season the export anomaly occurs, the stronger effect one might expect on the September minimum SIE. But working against this, is the overall decrease in export from March towards August (Fig. 2). However, even if there is some





re-growth in March, April and May, from increased spring export, the newly formed ice will be thin and have a thin snow cover, and therefore likely melt more easily and deform later the same season. The transition from winter and re-freezing, to summer and positive ice-albedo feedback, occurs gradually later in the year as one moves north, but melting will prevail over most of the Arctic Basin onwards from May (Markus et al., 2009). Williams et al. (2016) found that export anomalies in

the FS onwards from March contribute to anomalies in the September minimum that same year.

To estimate the direct effect of the spring ice export anomalies on the September SIE, we summarize the ice export anomalies onwards from March. We accumulate open water areas created by export and assume that they will contribute directly to open water areas in September the same year. The areas may not refreeze due to absorption of solar radiation (ice-

albedo feedback) and warmer air temperatures during summer, or they may form thin ice that melts later in the summer. The indirect additional effect of the ice-albedo feedback will be discussed in the next section. Note that we do not use any of the export anomalies before March in this approach.

Previous estimates of summer (June - September) ice export on the loss of perennial ice from 2005 to 2008 (Kwok and

Cunningham 2010) suggested only a small contribution. In contrast, our results show a stronger influence. The annual spring ice export was about 500,000 km² between 2011 and 2013 (Fig. 5). This is a 67 % increase compared to the spring export in the early 1980's, or an additional spring ice export of about 200,000 km². Over the same time-frame, the September SIE has decreased by about 2.0 million km², suggesting the recent increased spring export can directly explain around 10 % of the observed September SIE loss (Fig. 5). The particularly low September SIEs in 2007 and 2012 appear in part related to high

spring export during these years (Fig. 5). Similarly, during 2013 there was a decrease in spring ice export, and the September 2013 extent recovered.

While there seems to be a clear link between spring export and September SIE in recent years, this is less so further back in time. Nevertheless, the overall correlation for 1935 – 2014 between the two de-trended time-series is modest at r=-0.43

(Table 1).

One possible explanation for the missing response during earlier years lies in the sea ice dynamics and thickness variability in relation to regional wind forcing within the Arctic Basin. The thinner ice cover in more recent years deforms more easily and compacts given a convergent wind field, while in the past, the thicker ice cover could resist such wind forcing.  A

thinner ice cover is supported by thickness measurements in FS, which have shown thinning by about 1 m since 1992, while at the same time, the average age of the exported ice has decreased from 3 to 2 years (Hansen et al., 2014).





### 4.6 Spring Export and the Role of Positive Feedbacks

Any perturbations in Arctic sea ice cover may be further enhanced by positive feedback mechanisms during spring and summer. Of these, the ice-albedo feedback (Perovich et al., 2007), is the best known, but a thinner ice cover will also have a smaller resistance to wind forcing (Rampal et al., 2009) and deform more easily. Both processes amplify anomalies in a thinner ice cover and lead to formation of more open water areas. As noted by Markus et al. (2009) and Stroeve et al. (2014) early formations of open water areas are important in enhancing the ice-albedo feedback, allowing for more energy to be absorbed in the open water areas that in turn result in more basal and lateral ice melt. Since 1979 melt onset begins about 10 days earlier than it used to inside the Arctic Basin (Stroeve et al., 2014). Therefore, earlier melt onset combined with export anomalies between March and May together contribute towards less ice at the end of summer in September. In addition to the direct contribution of ice export of about 10 % on the September SIE found in section 4.5, we estimate the enhanced loss by ice-albedo feedback here.

A conservative estimate of the Arctic average ice-albedo feedback due to increased ice export can be made from the summer monthly mean incident solar irradiance observed at the Russian North Pole drift stations (Table 1, Björk and Söderkvist 2002), combined with a representative change in surface albedo, and the change in open water area (Perovich et al., 2007). To estimate the additional solar heating we first use the change in spring ice export from 1980 (Fig. 5. 1979-1982 average is 300,000 km²) to the present (2011-2013 average is 500,000 km²) to determine the increase in open water area inside the Arctic basin expected from increased ice export. Between these two time-periods, the additional ice area exported during spring (March – August) is thus 200,000 km². The monthly mean solar radiation at the North Pole is 33 W/m² in March, rises to 302 W/m² in June, and is 133 W/m² in August. The relevant change in surface albedo can be estimated as 0.53, i.e. the difference between a (melting) sea ice albedo of 0.6, and an open water albedo of 0.07 (Perovich et al., 2007).

Using the monthly mean ice export anomalies and monthly mean solar radiation, the additional March-August open water areas lead to an additional solar heating of $\sim 20 \times [10]^{20}$ J. This extra heat is enough to melt or prevent growth of 650 km³ of ice. Choosing a mean ice thickness of 1.5m (Zygmuntowska et al., 2014) for the ice (melted or prevented from forming) suggests an affected ice area of 435,000 km². The total contribution from increased spring ice export and the related ice-albedo feedback is thus estimated to be over 600,000 km² of ice. The mean September Arctic sea ice cover has decreased about 2.5 million km² between the two periods, and the combined ice export anomaly and ice-albedo feedback can thus explain 25 % of this loss. If the ice-albedo feedback is only applied onwards from June, the spring export anomalies still explains about 17 %.





### 4.7 Ice Export and other drivers of September SIE variability

Using coupled climate model simulations, Zhang (2015) identified the northward Atlantic heat transport, Pacific heat transport, and the spring Arctic Dipole (AD) as the main predictors of low-frequency variability of summer Arctic SIE. The study focused on variability longer than 30 years, and used a 3,600-year segment of the pre-industrial control simulation

from the Geophysical Fluid Dynamics Laboratory (GFDL) Coupled Model version 2.1 (CM2.1). The influence of oceanic heat transport is smaller for the year-to-year variability that we focus on in this paper, leaving the AD as one of the main causes of simulated summer Arctic SIE variability at the inter-annual time scale.

Using the same 3600-year long GFDL CM2.1 control simulation, we find that the simulated spring FS ice export is indeed

significantly correlated with the AD index (*r=0.63*, not shown). A larger correlation is found between the simulated AD index defined for April-July and the simulated export anomalies for those months. The link between the AD and FS spring export thus appears stronger in model simulations than for the available observations ($r_{AD-FS\ ice\ export} = 0.45$) between 1948 and 2014, but is confirmed to be present in a control simulation representing natural climate variability over 3600 years.

The simulations in Zhang (2015) suggested the AD to be one of the main drivers of low frequency summer Arctic SIE. Here we would like to estimate how much of this AD influence can be explained by FS ice export at the inter-annual time scale from observed correlations since 1979. In this period $r_{AD-Sept\ SIE} = -0.53$. We roughly estimate that 45 % of this correlation is caused indirectly by the FS spring export variability, because; $r = r_{AD-FS\ ice\ export} \times r_{FS\ ice\ export-Sept\ SIE} = 0.44 \times -0.54 = -0.24$, and (-0.24) / (-0.53) = 0.45. The FS spring export and the AD therefore have roughly equal

contributions to September SIE variability in addition to the common mechanism being stronger northerly geostrophic winds driving higher spring export. The additional direct AD influence on summer Arctic SIE is probably due to convergence inside the Arctic Basin, and atmospheric advection of heat and moisture (Graversen et al., 2011). The FS export influence independent from the AD is plainly export variability not driven by AD.

We also examined a forced historical simulation for the 20th century combined with a forced 21th century projection under the Coupled Model Intercomparison Project phase 3 (CMIP3) A1B scenario using the GFDL model. Such a simulation is forced with changes in all external forcings such as anthropogenic greenhouse gases and aerosols. In these simulations we found no significant trend in simulated spring FS ice export between 1979 and 2013. This is consistent with the 1935-2014 long-term trend being close to zero, and further suggests that the atmospherically driven FS ice export increase since 1979

(Fig. 3) is mostly due to natural variability.



### 4.8 Overall Effect of Spring Export

As discussed above in Section 4.6, removing the linear trends results in a $r_{FS\ ice\ export-Sept\ SIE} = -0.43$ for 1935-2014 (Table 1). This indicates that 18 % of the variance in the de-trended September SIE can be explained by the spring ice export. This shows clearly that the ice export influences the year-to-year variability (Fig. 5). However, the long-term trend since 1935 in September SIE cannot be explained by ice export variability, and must be caused by other factors, because there are no significant trends in annual, winter or spring ice export over the last 80 years.

More recently, there appears to be an influence of the FS spring ice export on September SIE onwards from 1980, i. e. a trend. As shown above, we estimate that this spring ice export increase directly explains about 10 % of the mid-September SIE ice loss, and when the ice-albedo effect is included, the response increases to 25 %. This is a physically based estimate, does not rely on any statistical assumptions, and indicates that the spring ice export over the last 35 years influences both the trend, as well as, the year-to-year variability.

For the recent ten years (2004-2014) when our ice export values are directly observed by SAR and the Arctic sea ice cover has been thinner and more responsive the detrended relationship is $r_{FS\ ice\ export-Sept\ SIE} = -0.74$ (Table 1). This suggests that the FS spring ice export explains about 55 % of the de-trended year-to-year variability of September SIE the last decade. Any additional contribution to the trend comes on top of this value. If we do not de-trend the time series, $r_{FS\ ice\ export-Sept\ SIE} = -0.72$. The additional explained variance is 17 %, quite similar to the physically based calculation above where we estimated the influence on the trend onwards from 1980 to be between 10-25 %.

### 5 Conclusions

A new and updated time-series of Fram Strait ice area export from 1935-2014 was presented in this study. The new time-series was constructed using high resolution radar satellite imagery of sea ice drift across 79°N from 2004 - 2014, regressed on the observed cross-strait surface pressure difference back to 1935. The overall mean annual export is 883,000 km², and there are no significant trends over this 80-year long time period. Winter export (September - February) carries about 60 % of the annual, while the spring export (March - August) carries the remaining 40 %.

The pressure difference from observed sea level pressure across the Fram Strait on Svalbard and Greenland directly explains 53 % of the variance in the observed ice export for 2004 - 2014. The best fit between ice drift and geostrophic winds results in a seasonal difference of ~3 cm/s, suggesting that the East Greenland Current, carrying a large part of the export, flows faster during winter and slower during summer, consistent with generally stronger large-scale wind forcing. The ice export





based on observed sea level pressure, including a seasonal variation in the underlying current, explains almost 80 % of the observed ice export variance.

While there is no long-term trend in export from 1935 to 2014, we do find positive and robust trends in sea ice area export over the last 35 years. This increase in export is created by stronger geostrophic winds, largely due to an observed increase in the surface pressure on Greenland, creating a positive trend of +6 % per decade for annual mean ice area export since 1979 (Fig. 3). The trend is mostly explained by the high trends for spring and summer months (March-August), when ice export has a robust trend of +11 % per decade.

The sea ice area export influence on the September SIE also seems to have increased in recent years, in part reflecting a thinner and more mobile sea ice cover. This influence is modest for the 80 years between 1935 and 2014, explaining about 18 % of the variance in September SIE. This level of influence is roughly similar to the exported anomalies during spring in sea-ice area directly from 1980 until today. This physical link also partially explains the correlation between the observed Arctic Dipole anomalies and the September SIE (Zhang 2015). This is simply the wind forcing (Arctic Dipole) driving the ice export, again leading to anomalies in September SIE.

Onwards from the 1990's, the influence of Fram Strait spring ice area export increases. Between 1993 and 2014, 22 % of the observed variance in mid-September Sea Ice Extent is explained by the spring ice export, increasing to 55 % for the last 10 years (Table 1). To reach such a level of influence, feedbacks are clearly necessary. Positive feedback mechanisms enhancing summer SIE anomalies are the ice-albedo feedback and increased deformation of thinner ice (Perovich et al., 2007; Rampal et al., 2009). During the last 10-20 years, the Arctic sea ice cover has decreased quite rapidly, and the contributions from natural variability and greenhouse gas forcing are still being debated. We calculated an important driver of Arctic sea ice variability for the last 80 years, and found that over this time scale there is no systematic increase in sea ice area exported southwards out of the Arctic Ocean in the Fram Strait. This is consistent with available historical simulations stating that we do not expect any systematic ice export change related to global warming (Langehaug et al., 2013). This is also consistent with studies stating that there is little systematic change in the Arctic large-scale circulation (Vihma 2014).

What we have found, is that there has been an increase in ice export over the last 35 years. This fact points to significant natural variability on multidecadal time-scales resembling the 60-80 year cycles found in Northern Hemisphere surface air temperature and temperature of inflowing Atlantic Water to the Barents Sea (Smedsrud et al., 2013). Consequently we speculate that there is potential for a partial recovery of the September SIE in the next decade or two, when, or if, the spring ice export decreases back to the long-term mean level of the last 80 years. However, the Arctic ice cover is now thinner and more mobile than before, and during the last three decades the September ice cover seems to have an increased coupling to the Fram Strait spring sea ice area export.



**Author contribution:** M. H. Halvorsen did most of the calculations of sea ice export based on the sea level pressure data, J. Stroeve calculated and plotted sea ice extent variations and helped re-focus the manuscript, R. Zhang contributed with analysing the Arctic Dipole time series and simulated sea ice export, and K. Kloster analysed the original SAR images and calculated monthly mean ice speed and export. L. H. Smedsrud prepared the manuscript with contributions from all co-authors and made most of the figures.

## Acknowledgements

Sea ice drift data was obtained from Kloster and Sandven (2014), where ScanSAR data was provided by Norwegian Space Centre and Kongsberg Satellite Service under the Norwegian-Canadian Radarsat agreements 2012 -2014. Observed pressure data are from the Norwegian- and Danish Meteorological Institute. The observed Arctic Dipole index is derived from the NCEP/NCAR reanalysis. M. H. Halvorsen was supported by the Geophysical Institute at the University of Bergen, Lars H. by the BASIC project in the Centre for Climate Dynamics (SKD) and the ice2ice project (ERC grant 610055) from the European Community's Seventh Framework Programme (FP7/2007-2013), and J. Stroeve by NASA Award NNX11AF44G. We would like to thank Tor Gammelsrød for helpful comments.





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



Table 1.   Correlations  over  time  between  Fram  Strait  Spring  Export  and  mid-September  Sea  Ice  Extent
($r_{FS\ ice\ export-Sept\ SIE}$)

|  | 1935-2014 | 1979-2014 | 1993-2014 | 2004-2014 |
|---|---|---|---|---|
| Export and extent | -0.23 | -0.55 | -0.49 | -0.72 |
| De-trended values | -0.43 | -0.39 | -0.47 | -0.74 |



**Figures and Captions:**

**Figure 1: The Arctic Ocean and surrounding shelf and land areas. The large black arrow shows location of Fram Strait, and the red circles positions of the meteorological stations with Sea Level Pressure observations. The 1935 to 2014 mean positions of the mid-month Sea Ice Extent is plotted for September (red) and March (black). The 1935-2014 mean annually exported sea ice area (883,000 km²) is illustrated by the polygon. The outer extent of the Arctic Ocean Domain is drawn using the yellow dashed line.**




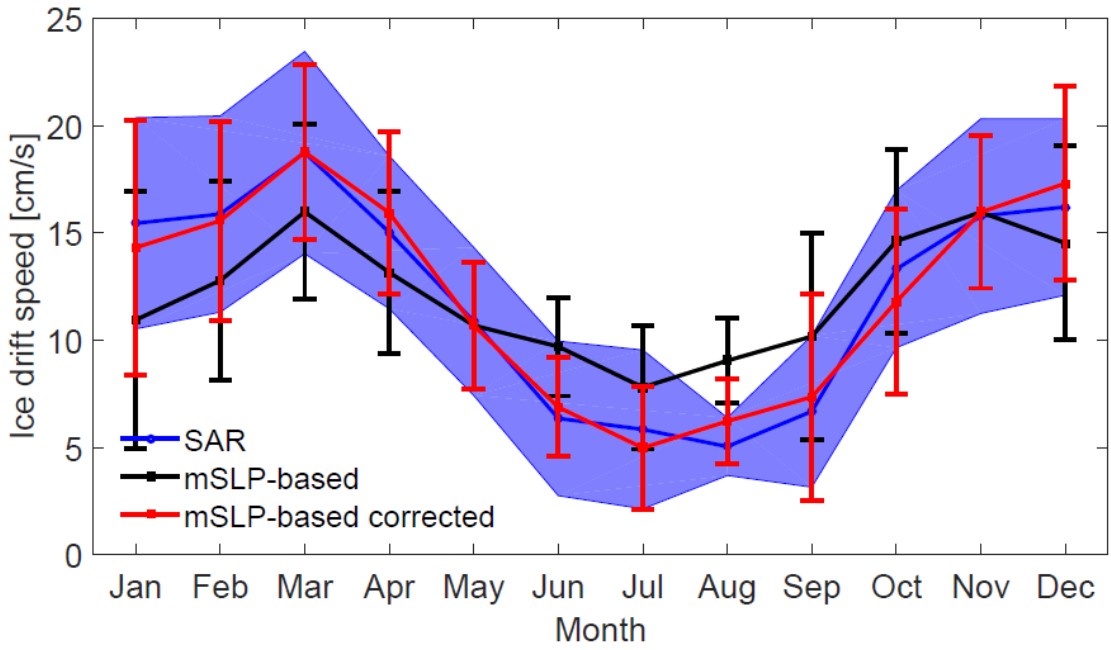

**Figure 2: Annual cycle of monthly mean southward ice drift speed in Fram Strait between 2004 and 2014. Observed ice drift speed**

5 **(SAR) are shown in blue, and our pressure based ice drift speed in black. The corrected ice drift speed is shown in red. Standard deviations of observed ice drift speed are shaded in purple, and of calculated ice drift speed as vertical, coloured lines.**





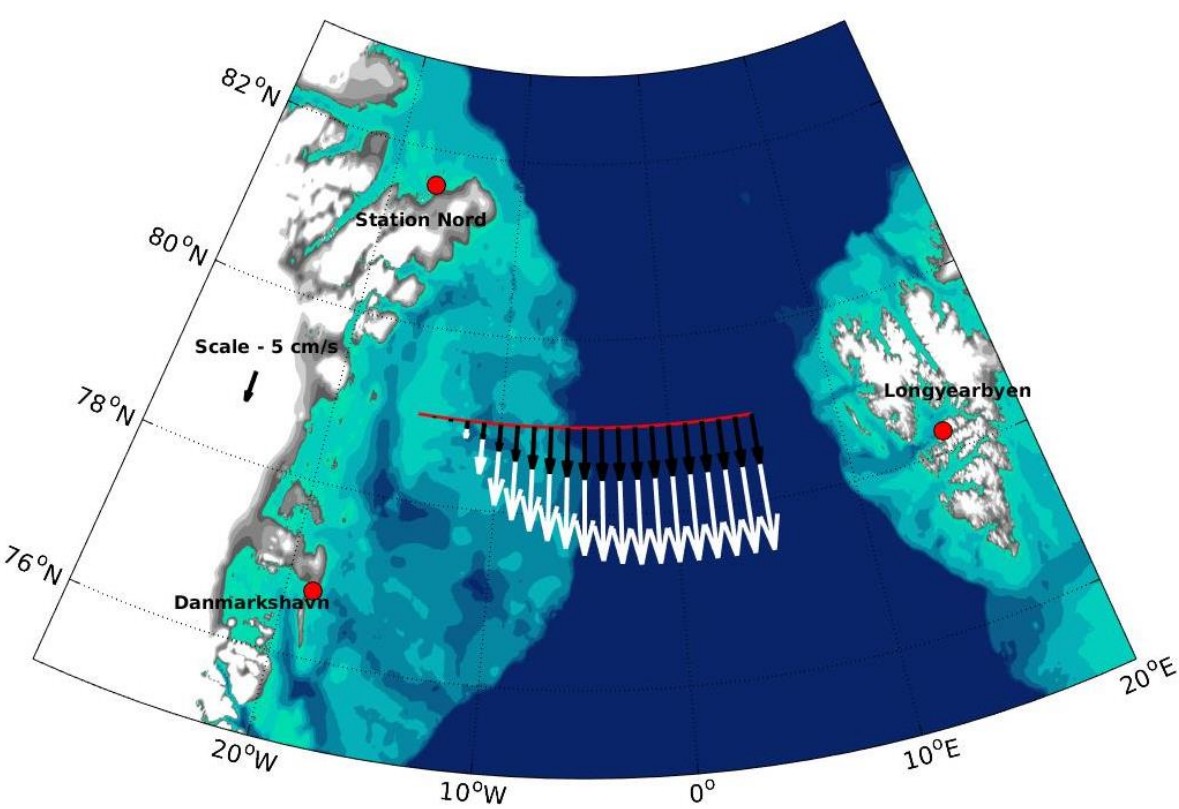

**Figure 3: The Fram Strait between Greenland (left) and Svalbard (right) with summer (black arrows) and winter (white arrows) mean sea ice drift speed. Southward ice drift across 79°N (red line) from February 2004 to December 2013 were interpolated to 1° bins based on SAR imagery. Summer speeds are June – September means, while winter speeds are December - March means. Shades of blue show ocean bathymetry in 100 m steps down to 500 m depth. Red circles show locations for surface pressure observations on Svalbard (Longyearbyen) and Greenland (Station Nord and Danmarkshavn). Pressure observations were interpolated between the Greenland stations to calculate the mean pressure gradient along 78.25°N. Before 1958 pressure observations from Danmarkshavn are lacking, so observations from Tasiilaq (65.60°N, 37.63°W) were used, further south along the Greenland coast (Fig. 1).**



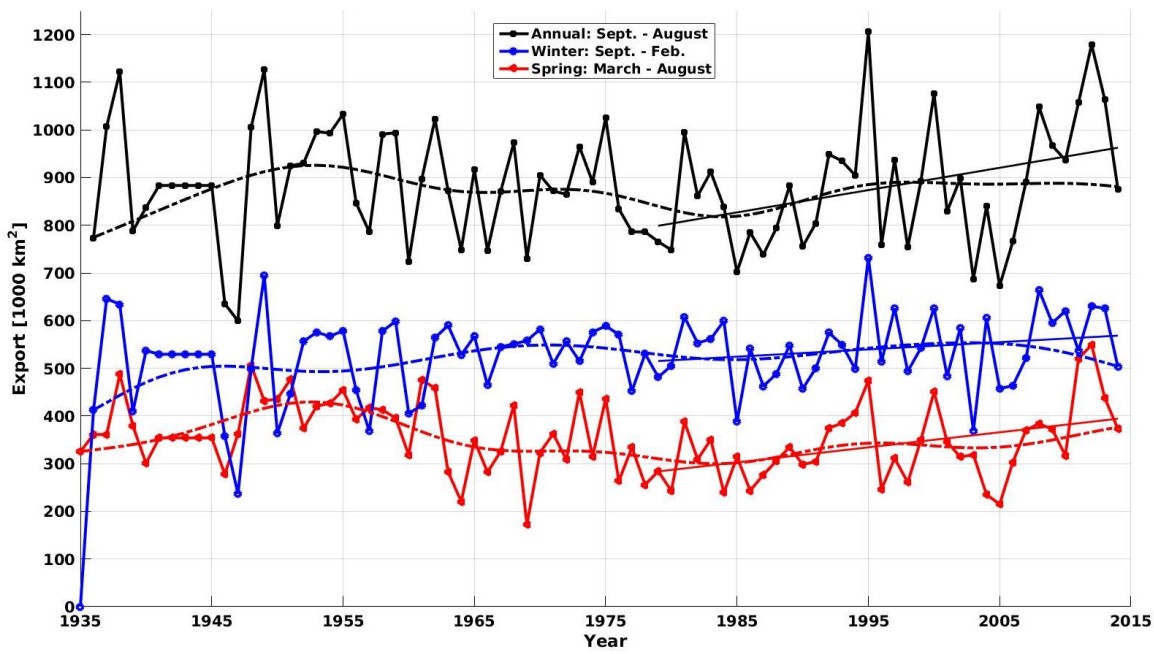

**Figure 4: Southward ice area export in Fram Strait. Ice export from 1935 - 2003 is based on the relationship between observed mean sea level pressure and observed ice export by SAR, and ice export from 2004 - 2014 are solely observations by SAR. Annual values (black) are averaged for September 1st through August 31st. Winter export is September 1st - February 28th. (blue) and Spring is March 1 st -  August 31 st. (red). Values are plotted half way through the respective period. Smoothed time-series are included produced by filtering with a 20 year cutoff 8-order Butterworth filter (dashed lines), and linear trends are plotted onwards from 1979. The long-term (1935-2014) trends are not included because they are not significantly different from zero.**



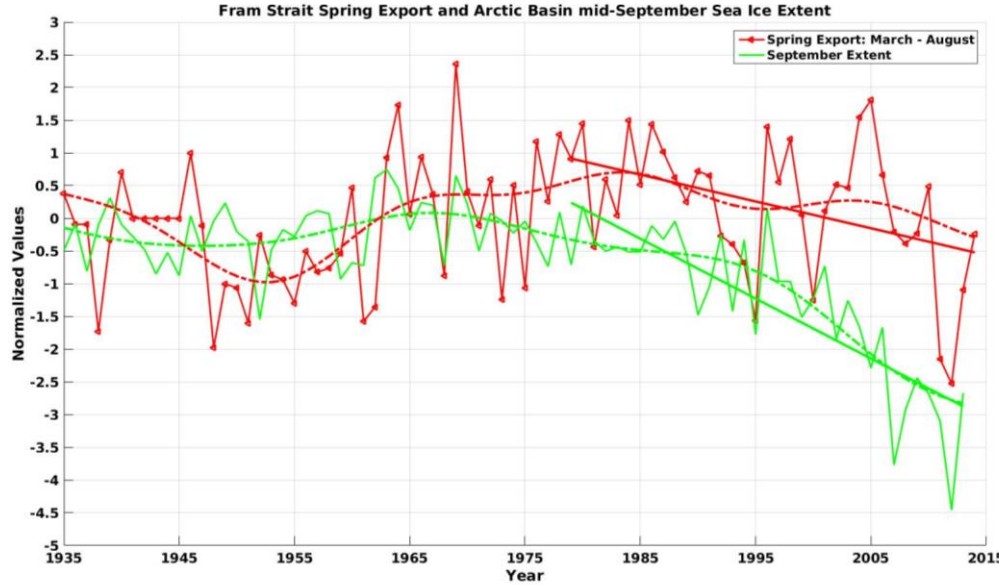

**Figure 5: Spring Fram Strait ice area export (red) and mid-September Arctic SIE (green). The ice export is averaged for March 1st through August 31st. Both time-series have been normalized by subtracting the mean and dividing with the standard deviation.**
5   **The ice export is here plotted with negative values as high southward export for easier comparison. Smoothed time-series are included produced by filtering with a 20 year cutoff 8-order Butterworth filter. The 1979 - 2014 trends in ice export and mid-September SIE is shown as solid straight lines. SIE values are obtained from Walsh et al., (2015).**



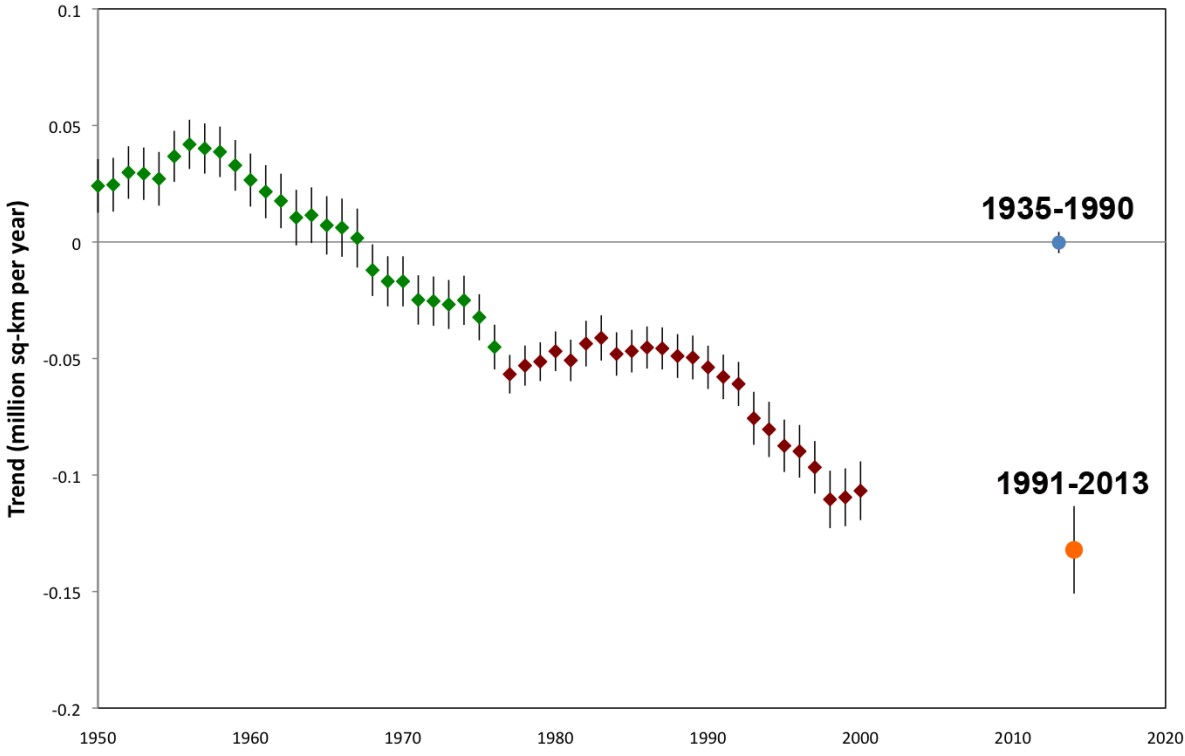

**Figure 6: Consecutive 30-year trends of mid-September SIE in the Arctic Basin. The first green symbol shows the 30 year trend between 1935 and 1964, and is plotted at the center year in 1950. The next value in 1951 shows the trend for 1936 to 1965, and so on. The last green symbol in 1975 is for the 1961 to 1990 trend. The red symbols show trends after 1990, ending with the 1984 to 2013 trend. The blue symbol shows that the 1935 to 1990 trend was zero, and the orange symbol shows the trend from 1991 to 2013. SIE values are obtained from Walsh et al., (2015).**



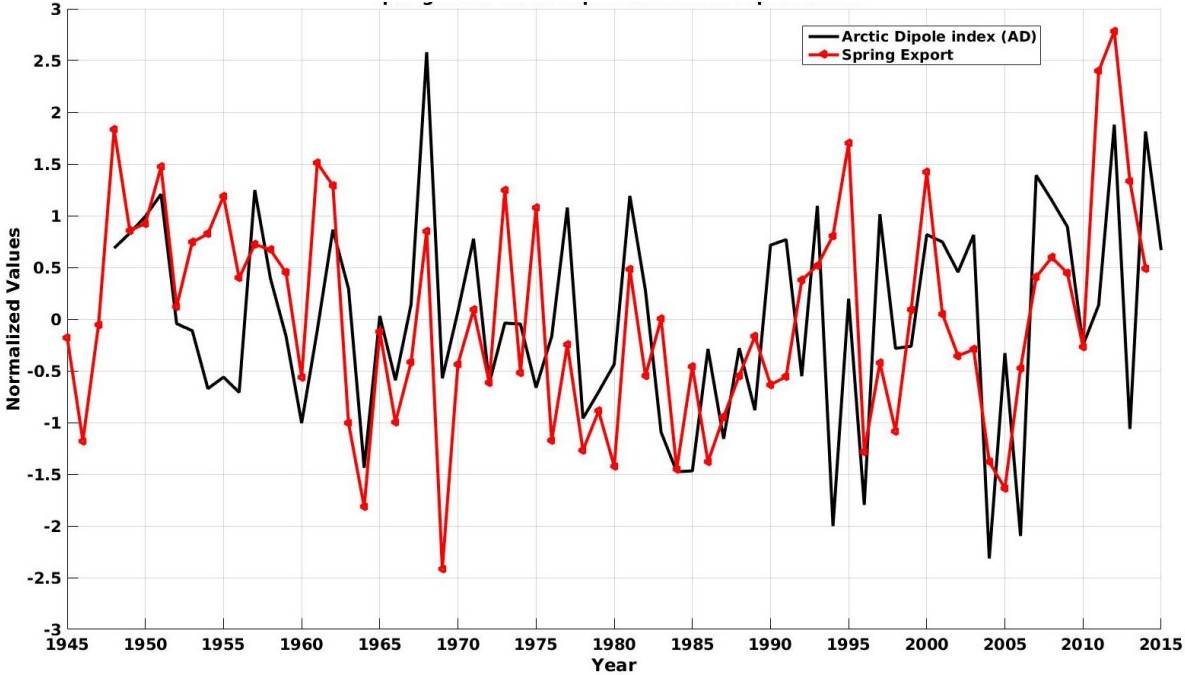

**Figure 7. : Fram Strait Spring export (March-August) and April-July Arctic Dipole (NCEP/NCAR reanalysis) anomalies from 1948 – 2014. Both time series are de-trended and normalized by their standard deviations, 0.817 million km² and 28.8 hPa, respectively. The correlation between them is $r_{AD-FS\ ice\ export} = 0.45$ for the entire period, and similar for 1979-2014**

5  **$(r_{AD-FS\ ice\ export} = 0.44)$.**