# Peer review of "Fram Strait sea ice export variability and September Arctic sea ice extent over the last 80 years"

_The Cryosphere, 2016_

## Referee Comment (RC1) · Anonymous Referee #1 · 3 Jul 2016

The manuscript discusses the Fram Strait sea ice area export over the last 80 years, i.e., from 1935 to 2014. Large variability but no longterm trend is found. However, during the last decade according to the presented time series ice area export increase. The authors, based on comparisons between spring ice export anomalies and summer minima, conclude that the increased ice export is is partially responsible for the accelerated decline in Arctic sea ice extent.

The variability and long term trends of the Arctic sea ice export and its connection to changes of the sea ice area within the Arctic Basin is an interesting and important topic. For the manuscript at hand I had many problems reviewing it because it (a) discusses and mixes very different datasets and methods, and (b) draws very bold

and far reaching conclusions based on quite simplified assumptions not taking the complexity of the coupled ocean-sea ice-atmosphere system enough into account:

- the authors construct a Fram Strait sea ice area flux proxy time series based on the across Strait air pressure gradient between Greenland and Svalbard. A regression between a high resolution SAR based ice area flux time series for 2004-2014 and the pressure gradient is performed. The regression coefficients (including a seasonal cycle adjustment) are used to reconstruct the sea ice area flux based on pressure observations alone. No sea ice observations are used before 2004 but only the air pressure. This fact was not initially clear to me as a reader from the methods section and I only understood it from the side note on page 9. Before the authors mention a new longterm sea ice extent time series (Walsh et al., 2015) but in the end they do not use it. This means that the time series before 2014 does not include any variability due to the changing sea ice area within Fram Strait. While Fram Strait is one of the areas in the Arctic with the smaller sea ice decrease during the satellite era it still shows a significant decrease. The time series presented here does not account for any such changes before 2004. These issues or other limits of the proxy time series are not discussed in the manuscript. On the contrary the authors never call it a proxy time series. These facts should be clearly mentioned already at the beginning of the document.

- the Walsh et al. sea ice extent time series covering the complete 1935-2014 period is used for comparisons between ice export and ice extent in the manuscript. For a revised version of the manuscript this dataset should be combined with the air pressure data to add some ice extent variability to the ice export time series, which should make it more realistic. It is unclear to me why this was not done. The Walsh et al. ice extent dataset is prominently introduced as a new and improved time series.

- the 2004 to 2014 part of the time series is based on ice area flux estimates based on manually derived sea ice drift from high resolution SAR imagery. This should give very good estimates of the ice area export. I still would have appreciated some discussion

of potential uncertainties due to the manual extraction by a human analyst or how they were mitigated. For example, were the number the spatial distribution of the manually derived ice drift vectors constant for the complete time series? It is my understanding that this time series was build up over many years. Can we assume that the quality is constant over time? The stated uncertainty of +-3 km for an individual ice drift vector is actual much higher than what I would have expected. The grid cell size of the SAR data is about 100m. Adding some uncertainty caused by geolocation variability and identifying the exact same point in two images I still would have expected an uncertainty on the order of 500m or better like for example reported for the Radarsat RGPS data.

- the authors then merge the air pressure proxy time series with the SAR based time series. The complete air pressure based ice export time series is not shown. In my opinion that should not be done. The two time series have very different error bars and characteristics. The air pressure gradient is the only information we have got to estimate the ice export before 1979 when the satellite data start. This is argument enough to use the air pressure as a proxy to derive and discuss the ice export variability. But again, it then also should be clearly stated what kind of time series is discussed in the manuscript. There is quite some focus on the 2004-2014 SAR dataset but the authors state themselves that this time period is too short to discuss significant trends. On page 7 the trends for the 1935-2014 air pressure time series alone are given and it is argued that these statistics are very similar to the merged time series. I would argue the other way around: use a consistent time series, i.e., the air pressure proxy ice export, for the complete period. This will avoid any biases, changes in statistics etc. due to the merging process in 2004.

- Figure 2 shows the similarity of the seasonal cycle between the adapted air pressure and SAR ice export time series. This is nice and shows good agreement but also differences for some months. For the reader it would be important to also see the two time series together for the complete 2004-2014 period. If the complete discussion in the manuscript would be changed to the air pressure only time series (see my last

point) the SAR derived time series could be added to Fig. 4 for comparison.

- The manuscript mentions that their ice export estimates for the last 30 years do not agree with estimates from passive microwave radiometers (e.g. Kwok et al., 2013). Actually, these satellite data based time series do not find a trend in ice export, which is opposite to the trend found here from the air pressure data. The authors attribute this difference to the low resolution of the satellite data and that it will not correctly track all ice in Fram Strait (p. 12). That is one possible explanation but the authors do not demonstrate this failure but hypothesis it. That is okay because the satellite data is not the topic of their study. But then the authors should be more critical also towards their own time series and list factors, which could explain the difference to the satellite data. For example: there is an increase in the across pressure gradient during the last 30 years. As this is the only data used in the proxy ice export time series presented here this directly results in a positive ice export trend. However, there are other factors, which influence Fram Strait ice export and could or have changed during the last decades and therefore counteract the increased pressure gradient:

(i) the ice area in Fram Strait (FS) shows a negative trend reducing the ice area export, which is not accounted for here.

(ii) the surface winds in FS are not only determined by the pressure gradient but have a strong contribution from thermal wind (THW) forcing (van Angelen et al., 2011). If the THW forcing would have been reduced during the last decades that would counteract the increased pressure gradient

(iii) the ice surface drag (surface roughness) could have changed, i.e., the atmosphere to ice energy transfer function can have changed. This could also be caused by a change of internal ice stress, i.e., how lose or compact the ice in FS is.

(iv) the ocean forcing can have changed

I don't know if these factors can explain the difference to the satellite ice export time
series but they should be discussed. Also in the summary it should be mentioned that all conclusion drawn here are based on the air pressure time series presented but that for other available ice export estimates one would get to complete opposite conclusions.

- In section 4 from 4.2 onwards the sea ice area export time series and the Walsh et al. sea ice extent time series are used to draw quite far reaching conclusion about the influence of the sea ice export increase they find on the recent decrease in Arctic sea ice area. They make the in my view oversimplified assumption that every spring ice export anomaly directly relates to a loss in ice area for the summer sea ice extent. There are many other factors which will influence this relationship, e.g., if the ice gets compressed or more spread out in the Arctic Basin and many more feedbacks the authors are well aware of. One would need a coupled Arctic regional climate model to make more robust conclusions about such relationship. I actually like such simplified speculations in the way of: "If we would assume the ice export anomalies to directly relate to anomalies in Arctic summer ice area this would mean . . ." But here they are presented as hard results and in a very broad way. I recommend to remove most of the discussion related to this in section 4 and concentrate on the new 80 year ice export time series at hand. Some of these hypothetical consequences can then be briefly mentioned at the end of the discussion.

The 80 year long air pressure based FS ice export time series by itself merits publication. Some information about the actual sea ice variability from the Welsh et al. dataset should be added. Errors and uncertainties have to be discussed more upfront and also in relation to other published but much shorter ice area export estimates. The mainly speculative discussion about consequences should be reduced and declared more clearly.

minor comments:

p7, l18: for 2011-2013 the export exceeds 1mil sq km.

[Figure]

p8, 3.2: is there a reason for choosing the period 1979-2014 beside that it maximises the the trend found in an on longterm average trend less time series?

p8, l19: in 2011 and 2012 the spring and winter exports are of similar magnitude but not in 2013 and 2014. Exports were on more similar magnitude during the 1940-50s. The reduction in seasonal cycle therefore is only temporarily.

p9, l3: I cannot see that Kwok, 2009 uses reanalysis data. They use satellite data.

p10, l13-14: In Fig 4 the 1995 export is larger than in 2012. That was also correctly stated before.

p11: see also Kwok et al., 2013 for a detailed discussion of AO and ice circulation.

p11: the purpose and conclusions from 4.3 regarding this manuscript remain a bit unclear to me. Better motivate or remove.

p13, l8-9: this is a very strong assumption (no feedbacks considered) and makes all conclusions based on this more hypothesis and speculations. Not a problem but should be clearly called that then and not presented the same way as the results based on the export time series. Could be more like an outlook section.

p13, l26-31: again speculative; the correlation of -0.43 is modest as you correctly say.

p14, 4.6: here you estimate the influence of one feedback. But there are many others. See e.g. the influence of ice convergence along the CAA contributing to the 2012 minimum. As a fully coupled system I am not sure one can simply separate feedbacks and sum them up again in the end. All feedbacks will interact with each other, there are many non linear responses. A coupled GCM would be a better approach to evaluate this.

p15, 4.7: here you look at a GCM but only in relation to AD. Does the GFDL model show high correlations between spring export and summer ice area minima?

---

## Referee Comment (RC2) · Anonymous Referee #2 · 19 Jul 2016

This paper attempts to extend the time series of Fram Strait (FS) ice export back to 1935. My primary concern with this paper is the accumulated errors in their regression of ice velocities going back to 1935. Given these uncertainties, I don't think they can make any definitive conclusions based on the extrapolated time series. Details on this concern and other comments are provided below. I suggest rejection of this paper.

1) Standard error about the regression line for equation 1.

The authors state a standard error of the regression line of 3.4 cm/s. Ice velocities are typically 12 cm/s. The error adds up to an ice export uncertainty of +/- 250000 kmˆ2, which is also the variance about the mean of 883000 kmˆ2. Given this uncertainty, it is hard to trust any conclusions drawn on their extrapolated time series, which is

foundation of this paper.

2) Fram Strait SLP Gradient

I think the linear interpolation to estimate pressure to 78N after 1958 is probably OK since the stations are close, but prior to 1958, the southern station may be too far? The authors need to substantiate the use of the 3 weather stations on Greenland to interpolate SLP at 78N and estimate the across strait pressure gradient.

One way to do this is to compare the estimated SLP at 78N based on the regression from Nord to Danmarshavn, and Nord to Tasillaq during a period when they have data from all 3 stations.

Equation 1 should also be evaluated based on the 2 estimates of dp/dx across the strait to see how much difference the use of the different stations make.

3) The authors should cite Hilmer and Jung, 2000 "Evidence for a recent change in the link between the North Atlantic Oscillation and Arctic sea ice export", in any discussion of Fram Strait ice flux. I think this is the definitive paper on the topic. Given that HJ cover the period going back to 1958, and many of the authors own papers discuss the period after this to the present. This paper would really have to substantiate their estimates for export prior to 1958 to make an acceptable contribution to the literature.

---

## Author Comment (AC1) · 8 Aug 2016

Thank you for your review, although it was rather short and not so positive.

Needless to say we were rather disappointed with your suggestion to reject the paper, and our response below substantiates our view on the issues raised. We include the original comments in bold font below. We appreciate the reviewer pointing out the Hilmer and Jung (2000) paper which we had overlooked.

Our results are consistent with those presented by Hilmer and Jung (2000), yet our study improves upon their results by 1) evaluating the long-term trend over a longer time-period (1935-2014) than they considered (1958-1997); 2) using station observations instead of NCEP reanalysis data, which have known problems; 3) analysis of the more Arctic relevant AO instead of the NAO; and 4) we find that the AD is a better index for explaining Fram Strait ice area Export than AO and NAO. Note that Hilmer and Jung (2000) only presented winter data (DJFM), but we present data for all months through the year, and we find substantial changes between winter and spring (Figure 4) over time as well. In their discussion Hilmer and Jung also state that "it cannot be decided wether NAO and Arctic Sea ice export are significantly related in a long-term context", and continued; "This question might be addressed by analyzing historical SLP data ". This is indeed what we have done in our paper here.

**This paper attempts to extend the time series of Fram Strait (FS) ice export back to 1935. My primary concern with this paper is the accumulated errors in their regression of ice velocities going back to 1935. Given these uncertainties, I don't think they can make any definitive conclusions based on the extrapolated time series. Details on this concern and other comments are provided below. I suggest rejection of this paper.**

The reviewer seems to have not quite understood the methodology. Our ice export time series is not based on extrapolation, nor on simulations. Rather, it is based on observations of surface pressure. A large number of papers find that ice drift is proportional to the geostrophic wind (this is basic physics), and there is no reason to expect the uncertainties in the calculated monthly means should accumulate. On the contrary will the uncertainties in the monthly means (based on 30 daily observations of pressure, and 10 values of the SAR derived ice drift) be further reduced when they are averaged into the seasonal means as described below.

**1) Standard error about the regression line for equation 1. The authors state a standard error of the regression line of 3.4 cm/s. Ice velocities are typically 12 cm/s. The error adds up to an ice export uncertainty of +/- 250000 km$^2$, which is also the variance about the mean of 883000 km$^2$. Given this uncertainty, it is hard to trust any conclusions drawn on their extrapolated time series, which is**

**foundation of this paper.**

The standard error is a statistical estimate of uncertainty, and describes the scatter around the regression line. The scatter is caused by the other factors influencing sea ice drift other than the geostrophic wind, and will be close to normally distributed around the regression line. The method is the same as used in Smedsrud et al (2011), but with 5 years of extended data. The uncertainty will further decrease when averaging into seasons is performed, because some months have slightly higher speed than the regression predicts, and some will have lower. So it is not correct to "add up" the uncertainty as suggested by the reviewer here.

Such a level of uncertainty is common in geophysics. When regression is used for non-physical relationships one should be very careful, but here the regression confirms first-order physics that the ice speed is proportional to the geostrophic wind. A correlation of 0.73 is very high. The correlation further increased to 0.88 after the correction on the East Greenland Current was done, reflecting that the SLP gradient is the most useful estimate of Fram Strait Export before detailed satellite imagery became available. We have explained clearly how the observations have been analyzed, and no formal errors in our data analysis have been suggested. The uncertainty and method is similar to previous published estimates, and seasonal mean values that are mostly used in our paper will have a lower uncertainty than the standard error derived from monthly mean values.

**2) Fram Strait SLP Gradient I think the linear interpolation to estimate pressure to 78N after 1958 is probably OK since the stations are close, but prior to 1958, the southern station may be too far? The authors need to substantiate the use of the 3 weather stations on Greenland to interpolate SLP at 78N and estimate the across strait pressure gradient. One way to do this is to compare the estimated SLP at 78N based on the regression from Nord to Danmarkshavn, and Nord to Tasillaq during a period when they have data from all 3 stations.**

**Equation 1 should also be evaluated based on the 2 estimates of dp/dx across the strait to see how much difference the use of the different stations make.**

We did perform correlations between the stations as described on page 4, line 11-18. There is relative lower but significant correlation (r=0.77 instead of r =0.93) between Nord and Tasiilaq, and there are no other alternative observations available prior to 1958, so this is the best data we have. The SLP pattern tends to follow the Greenland coast quite well (Fig. 4c in Hilmer Jung (2000) for example), and what matters here is the East-west SLP gradient, which should be robust.

**3) The authors should cite Hilmer and Jung, 2000 "Evidence for a recent change in the link between the North Atlantic Oscillation and Arctic sea ice export", in any discussion of Fram Strait ice flux. I think this is the definitive paper on the topic. Given that HJ cover the period going back to 1958, and many of the authors own papers discuss the period after this to the present. This paper would really have to substantiate their estimates for export prior to 1958 to make an acceptable contribution to the literature.**

Thank you for pointing out this paper, we will cite it in an updated version. We find that some of our conclusions are consistent with their results. Prior to 1978 Hilmer and Jung (2000) used simulations with quite a coarse resolution numerical model driven by another set of simulations (NCEP reanalysis) that are now known to have several issues in the Arctic. We therefore have more confidence in our own results for the early time period, as they are directly based on observations. Note that both the "missing" link between NAO and winter export, and a (not discussed) long term trend 1958 – 1997 is qualitatively consistent with our results. We used the AO index in our discussion (page 11) instead of the NAO, as it is a better index for the Arctic large scale atmospheric circulation and is also highly correlated with the NAO.

---

## Editor Comment (EC1) · J. Hutchings (Editor) · 10 Aug 2016

Dear Lars and co-authors,

As you have learnt, all the reviews and comments are in for your paper "Fram Strait sea ice export variability and September Arctic sea ice extent over the last 80 years". At this stage I would like you to provide a response to reviewer 1 and outline changes you will make to the paper in response to all comments of all reviewers. Thank you for the response to reviewer 2.

Both reviewers express concerns about the presentation of uncertainty in your manuscript. In response to reviewer 2 could you please discuss how the uncertainty

in ice drift estimate impacts the area flux estimate. Yes, I agree the variance is not the same as the error estimate, however it is very hard from your current text and figures to gauge the signal to noise ratio. In the context of the trends and whether you can identify periods of variability that are longer than the interannual variability this variance does impact the length of time series you need to draw conclusions. Your manuscript could be clarified in this respect, which would of course make the paper more readable and accessible. For example, you discuss in detail the difference in trends between various different analyses, but do not put this into the context of if these differences are statistically insignificant. You do discuss the point that the last 30 years is impacted by several high export years at the end of the time series, and I think you can do a better job of putting this into context. The fact that your SLP based export estimate differs from previous work needs to be considered in the context of no particular record being the truth and all having errors that are not well defined.

There are several previous works linking the Fram Strait export or the Arctic sea ice pack state and ice motion to the Dipole Anomaly. Please consider the work of Jia Wang's group for example.

Wu, B., Wang, J., & Walsh, J. E. (2006). Dipole anomaly in the winter Arctic atmosphere and its association with sea ice motion. Journal of Climate, 19(2), 210-225.

Watanabe, E., Wang, J., Sumi, A., & Hasumi, H. (2006). Arctic dipole anomaly and its contribution to sea ice export from the Arctic Ocean in the 20th century. Geophysical research letters, 33(23).

Wang, J., Zhang, J., Watanabe, E., Ikeda, M., Mizobata, K., Walsh, J. E., ... & Wu, B. (2009). Is the Dipole Anomaly a major driver to record lows in Arctic summer sea ice extent?. Geophysical Research Letters, 36(5).

Your choice of calling March to August Spring is a little unusual in my mind.

I understand that you choose this time period based on the assumption all ice that

grows in open water created by the export between these times will melt out in summer. This is a highly simplified model and does not account for ridging, but then again the albedo feedback will amplify the melt so you are not really looking at export from March-August as a linear indicator of open water in summer. Did you choose to split the time series periods in March as this is the 6 month split that gives you the best fit of the export to end of summer ice extent? I am having trouble wrapping my head around how the over 80% variance in ice export explained by cross strait pressure gradient and assumed seasonal cycle on ocean currents(at a time when the time series is also experiencing larger export), and the 22-55% covariance of the export proxy and following September sea ice extent allow you to make strong statements about causality. While the proxy record is defendable, I am not sure what contribution export has to ice extent based on the correlations. Does the proxy perform as well earlier in the time series, how much of the reduction in correlation is due to decreased covariance of the three station pressure with ice drift? In fact, the shorter time period is influenced by specific high export years, and you show that the last 7 years of the record are where this happens and influences trends. If you were to chose a similar short time series bracketing the years identified by for example Son Ngheim et al. (2007) as high transpolar ice drift and export (e.g. 2005-2007) would you get increased correlation based on this particular event? It does not look so from my quick scan of your figures, and the ice export at Fram Strait lagged (by a year) the transpolar drift event that the 2007 minimum was related to. It appears to me that the only time when the export explained a significant portion of the September ice extent has been in recent years (2011-2014). Is this the case?

I agree with reviewer 1 that you should tightening your manuscript to not overstate results.

Specific Points

Abstract line 18 and 20: Missing squared from your area dimensions. Also at page 12, lines 20 and 21. Check throughout please.
line 32 "FShas" -> "FS has"

line 25-26 "should be considerably more accurate that 10%". Did you not actually estimate this? I think you just need to reconsider your grammar here.

page 5 line 16-19 You have noted an increased ice drift in winter. Echoing reviewer 1, there is increased open water in summer and potentially changes to surface roughness of the ice. This will impact stress transfer between the wind and ocean, and increased wind stress transfer to the ocean might also lead to increased currents. This is an example of speculative discussion where you could strengthen the manuscript by focussing on your key result (the time series) and a more rounded acknowledgement of its limitations.

page 16 line 25: There is a missing word

Please check all your references are listed. I could not find Krumpen et al. (2016) for example.

Looking forward to you response, Jenny

⸻

---

## Author Comment (AC2) · 15 Aug 2016

Thank you for your very thorough and helpful review. The suggestions for changes have improved the manuscript, and on most issues raised a change has been implemented. Our response below substantiates our view on the issues raised, and explains our view for the few concerns raised that we do not think impact the conclusions presented in this paper. We include the original comments in **bold** font below.

The two major issues raised by the reviewer are (1) why we did not use the ice concentrations from the Walsh et al (2015) data set in the estimates of ice export prior to 2004, and (2) that the discussion is rather speculative. In response, we have looked more into the pre 2004 ice concentration data (see aFigure 1 below). Preliminary analysis of this

[Figure]

data shows that while there is increased variability, there is little long-term change that would systematically change the results we present in the paper and our discussion of trends. We also improve our description of the links between ice export and September ice extent. Our results are consistent with coupled GFDL climate model simulations, and also the recently published Williams et al (2016) paper that finds Fram Strait export to be the most important predictor for September SIE for 1993 – 2014, confirming our results are less speculative than the reviewer assumed.

**The manuscript discusses the Fram Strait sea ice area export over the last 80 years, i.e., from 1935 to 2014. Large variability but no longterm trend is found. However, during the last decade according to the presented time series, ice area export increase. The authors, based on comparisons between spring ice export anomalies and summer minima, conclude that the increased ice export is partially responsible for the accelerated decline in Arctic sea ice extent. The variability and long term trends of the Arctic sea ice export and its connection to changes of the sea ice area within the Arctic Basin is an interesting and important topic.**

**For the manuscript at hand I had many problems reviewing it because it (a) discusses and mixes very different datasets and methods, and (b) draws very bold and far reaching conclusions based on quite simplified assumptions not taking the complexity of the coupled ocean-sea ice-atmosphere system enough into account:**

**- the authors construct a Fram Strait sea ice area flux proxy time series based on the across Strait air pressure gradient between Greenland and Svalbard. A regression between a high resolution SAR based ice area flux time series for 2004-2014 and the pressure gradient is performed. The regression coefficients (including a seasonal cycle adjustment) are used to reconstruct the sea ice area flux based on pressure observations alone. No sea ice observations are used before 2004 but only the air pressure. This fact was not initially clear to me**

**as a reader from the methods section and I only understood it from the side note on page 9. Before the authors mention a new longterm sea ice extent time series (Walsh et al., 2015) but in the end they do not use it. This means that the time series before 2014 does not include any variability due to the changing sea ice area within Fram Strait. While Fram Strait is one of the areas in the Arctic with the smaller sea ice decrease during the satellite era it still shows a significant decrease. The time series presented here does not account for any such changes before 2004. These issues or other limits of the proxy time series are not discussed in the manuscript. On the contrary the authors never call it a proxy time series. These facts should be clearly mentioned already at the beginning of the document.**

We have tried to explain our methods as clear as possible, and start section 2 describing the ice drift observations from 2004-2014. We then continue with the Sea Level Pressure observations from 1935 – 2014. We describe how we blended these in section 2.3. While this is not a standard method we have clearly stated what we did. The term "proxy" is usually used for paleo observations like different organisms found in sediment cores that in some way reflect for example surface temperature. The physical relationship between SLP and ice drift is strong and qualitatively very different to this use of the word. We thus used the term "mSLP based" to describe our ice export estimates prior to 2004. This term was used in section 3.1 for example.

Fram Strait ice concentration change has been, and remains, small. Smedsrud (2011) found a small decrease in sea ice concentration across 79N for the period 1979 to 2009 of $-1.3\%$ per decade, and we have looked more into this in the new version using the Walsh et al (2015) data. For example, Figure 1 shows the mean (15W $-$ 5 E) ice concentration along 79N. While there is considerable year-to-year variability, there is little long-term change.

**- the Walsh et al. sea ice extent time series covering the complete 1935-2014 period is used for comparisons between ice export and ice extent in the**

**manuscript. For a revised version of the manuscript this dataset should be combined with the air pressure data to add some ice extent variability to the ice export time series, which should make it more realistic. It is unclear to me why this was not done. The Walsh et al. ice extent dataset is prominently introduced as a new and improved time series.**

Yes - we used the new Walsh et al (2015) data set primarily to evaluate effects of sea ice export, because we wanted to investigate September SIE variability in relation to ice export. It is not straight-forward to combine it with the SLP observations to make a new and more 'realistic' ice export because it only provides a mid-month ice concentration field. For 2004 – 2014 we use ice concentration for the same days as the SAR imagery. While using the mid-month extent for 1938 – 2004 is worth looking into, it would create a "shift" pre and post 2004, and the reviewer also states that it would be better to use the "mSLP based" ice export series for the entire 1938 – 2014 period, so these two suggestions cannot both be implemented. We instead analyze the long-term changes in ice concentration and extent from 1938 – 2004/2014, and discuss the consequences of this change in the new version as shown in the figure.

**- the 2004 to 2014 part of the time series is based on ice area flux estimates based on manually derived sea ice drift from high resolution SAR imagery. This should give very good estimates of the ice area export. I still would have appreciated some discussion of potential uncertainties due to the manual extraction by a human analyst or how they were mitigated. For example, were the number and the spatial distribution of the manually derived ice drift vectors constant for the complete time series? It is my understanding that this time series was build up over many years. Can we assume that the quality is constant over time? The stated uncertainty of +-3 km for an individual ice drift vector is actual much higher than what I would have expected. The grid cell size of the SAR data is about 100m. Adding some uncertainty caused by geolocation variability and identifying the exact same point in two images I still would have expected an uncertainty on the**

**order of 500m or better like for example reported for the Radarsat RGPS data.**

The SAR time series has images every three days for the 2004 – 2014 period, and have been manually derived by the same person, Kjell Kloster, for all that time. The details are described in a report; Kloster and Sandven (2014). Although it is manually derived, having the same person doing it should lead to a constant quality over time. An independent test of a SAR image pair by the University of Tasmania (Heil, personal comm. 2012) showed that a computer image tracker could re-produce about 60% of the velocity vectors, but gave basically the same vectors for those that were picked up. We have added a better description in the paper now, and give more details below.

- SAR ice displacement is based on comparing two 3-day interval images, each with 300 -500m pixels. These are resampled from images with 50 -100m pixels in order to reduce the SAR specle noise, thus greatly improving feature recognition and to ensure that the same feature is found on each image and correctly tracked over the interval. Gridding to 2km accuracy is done using the known satellite orbit and instrument parameters in addition to one reference point, and has varied for the different SAR data suppliers used between 2004 and 2014. Drifting platforms with GPS were sporadically present in the Strait and used to check the SAR drift accuracy, indicating errors of about 3%. Similar errors were found for the comparison with several drifting IAPB buoys.

The ice concentration is measured by passive microwave (SSMI, AMSR) that has an estimated uncertainty of about +-3%, resulting in a flux uncertainty of about +-5% in the 3-day fluxes. Neglecting any biases, adding 10 values to get the monthly flux would decrease the uncertainty by a factor of three. An uncertainty estimate of the monthly fluxes of about +- 5% is therefore also conservative. For the seasonal 6-monthly means mostly discussed here, the estimated uncertainty further reduces as the individual months are truly independent. A total seasonal area flux of about 500.000 $km^2$ should thus have an estimated uncertainty smaller than +-11.2000 $km^2$.

**- the authors then merge the air pressure proxy time series with the SAR based**

**time series. The complete air pressure based ice export time series is not shown. In my opinion that should not be done. The two time series have very different error bars and characteristics. The air pressure gradient is the only information we have got to estimate the ice export before 1979 when the satellite data start. This is argument enough to use the air pressure as a proxy to derive and discuss the ice export variability.**

**But again, it then also should be clearly stated what kind of time series is discussed in the manuscript. There is quite some focus on the 2004-2014 SAR dataset but the authors state themselves that this time period is too short to discuss significant trends. On page 7 the trends for the 1935-2014 air pressure time series alone are given and it is argued that these statistics are very similar to the merged time series. I would argue the other way around: use a consistent time series, i.e., the air pressure proxy ice export, for the complete period. This will avoid any biases, changes in statistics etc. due to the merging process in 2004.**

We agree that this is an important question, and it is exactly why we discussed this merging in three different paragraphs (Page 7, line 23 – 29, Page 8, line 13 – 33). We did however end up on the opposite conclusion that the best thing was to present the "best possible" merged time series. The trends would be very similar if we should follow this suggestion and plot that in Fig. 2 instead. We will re-consider this in the new version, but no significant changes should occur. Note that the above suggestion of using ice concentrations from Walsh et al (2015) for at least 1938 – 1979, would lead to another "shift". Using the pressure based ice export as suggested here actually thus suggest not to use the Walsh et al (2015) data.

**Figure 2 shows the similarity of the seasonal cycle between the adapted air pressure and SAR ice export time series. This is nice and shows good agreement but also differences for some months. For the reader it would be important to also see the two time series together for the complete 2004-2014 period. If the complete discussion in the manuscript would be changed to the air pressure only**

**time series (see my last point) the SAR derived time series could be added to Fig. 4 for comparison.**

We understand the importance of checking the agreement between the mSLP based and SAR based values. Fig. 4 in Smedsrud (2011) shows such a comparison for 2004 – 2010. The updated values are similar, and we found no particular reason to include them as a separate figure here. From visual inspection of Fig. 4 here it should be clear that there are no significant differences in the merged values on either side of 2004.

**- The manuscript mentions that their ice export estimates for the last 30 years do not agree with estimates from passive microwave radiometers (e.g. Kwok et al., 2013). Actually, these satellite data based time series do not find a trend in ice export, which is opposite to the trend found here from the air pressure data. The authors attribute this difference to the low resolution of the satellite data and that it will not correctly track all ice in Fram Strait (p. 12). That is one possible explanation but the authors do not demonstrate this failure but hypothesize it. That is okay because the satellite data is not the topic of their study. But then the authors should be more critical also towards their own time series and list factors, which could explain the difference to the satellite data. For example: there is an increase in the across pressure gradient during the last 30 years. As this is the only data used in the proxy ice export time series presented here this directly results in a positive ice export trend.**

**However, there are other factors, which influence Fram Strait ice export and could or have changed during the last decades and therefore counteract the increased pressure gradient:**

**(i) the ice area in Fram Strait (FS) shows a negative trend reducing the ice area export, which is not accounted for here.**

**(ii) the surface winds in FS are not only determined by the pressure gradient but have a strong contribution from thermal wind (THW) forcing (van Angelen et al.,**

2011). If the THW forcing would have been reduced during the last decades that would counteract the increased pressure gradient

(iii) the ice surface drag (surface roughness) could have changed, i.e., the atmosphere to ice energy transfer function can have changed. This could also be caused by a change of internal ice stress, i.e., how lose or compact the ice in FS is.

(iv) the ocean forcing can have changed

I don't know if these factors can explain the difference to the satellite ice export time series but they should be discussed. Also in the summary it should be mentioned that all conclusion drawn here are based on the air pressure time series presented but that for other available ice export estimates one would get to complete opposite conclusions.

The reviewer states an important point, and we have not tried to "minimize" the sea ice export variability not related to SLP. We agree that there are a number of physical parameters that could have changed over these 80 years, and we have added a better discussion of these points in the new version. All points i) – iv) are valid, and will be included. We quickly summarized the "non SLP related" variability to be 20% on page 6 (line 3 – 6), but we will extend this text.

However – we have little or no observations of these parameters for the time frame from 1935 – 1979. An exception is i) that may now be estimated using the Walsh et al (2015) data. For ii) we agree that the stronger thermal wind forcing during winter (van Angelen et al. 2011) is another explanation for the larger export during winter than estimated by the mSLP. We discussed this seasonal difference and attributed it to a stronger East Greenland Current (EGC, page 5 line 10 – 28). It is also consistent with a stronger thermal wind, and note that the simulations of van Angelen et al. (2011) did not include an ocean model, so the thermal wind could well explained the stronger current during winter.

For iii) the main influence is probably due to thickness change, and it is likely that before 2004 ice was thicker and moved less effectively for a given mSLP gradient as found by Kwok et al (2013). This would lead to smaller values of ice export prior to 2004, and would thus increase the trend onwards from 1979. Our results remain different from for example Fig. 7 of Kwok et al (2013) that finds a positive trend for summer ice area export (June – September), but not for the annual values.

The main difference from Kwok et al (2013) is the 2004 – 2014 time period when we have higher export values. In this period we use the observed passive microwave sea ice concentration. This is in short why we wanted to present the "best possible" time series and not the "mSLP based" time series as suggested above.

**- In section 4 from 4.2 onwards the sea ice area export time series and the Walsh et al. sea ice extent time series are used to draw quite far reaching conclusion about the influence of the sea ice export increase they find on the recent decrease in Arctic sea ice area. They make the in my view oversimplified assumption that every spring ice export anomaly directly relates to a loss in ice area for the summer sea ice extent. There are many other factors which will influence this relationship, e.g., if the ice gets compressed or more spread out in the Arctic Basin and many more feedbacks the authors are well aware of. One would need a coupled Arctic regional climate model to make more robust conclusions about such relationship. I actually like such simplified speculations in the way of: "If we would assume the ice export anomalies to directly relate to anomalies in Arctic summer ice area this would mean . . ." But here they are presented as hard results and in a very broad way. I recommend to remove most of the discussion related to this in section 4 and concentrate on the new 80 year ice export time series at hand. Some of these hypothetical consequences can then be briefly mentioned at the end of the discussion.**

We understand the reviewers point. Specified simulations using a regional climate model could be performed for another way of estimating the effects of the sea ice export variability. Such model simulations are complicated, and have not been performed. Using a dynamical sea ice drift model Williams et al (2016) have actually performed experiments using coastal divergence and Fram Strait export, and find a similar level of influence on the September SIE. We are indeed aware of many other factors influencing September SIE variability, and only state that between 18% - 22% is caused by the export, apart from in the last 10 years. Our understanding is also based on the long control run from the coupled GFDL model. In a previous version of this paper (Halvorsen at al 2015, The Cryospere Discussions) these model results were included in more detail, and backed up our understanding. They were subsequently removed due to a previous reviewer's suggestion. We will add some of this text back and thus come out as less speculative in the next version.

**The 80 year long air pressure based FS ice export time series by itself merits publication. Some information about the actual sea ice variability from the Walsh et al. dataset should be added. Errors and uncertainties have to be discussed more upfront and also in relation to other published but much shorter ice area export estimates. The mainly speculative discussion about consequences should be reduced and declared more clearly.**

Thank you for your interest in the export itself. We agree, but also found that more people are interested in the export if the plausible consequences are also discussed. This is what we attempted to do here.

**Minor comments:**

**p7, l18: for 2011-2013 the export exceeds 1mil sq km.**

Corrected.

**p8, 3.2: is there a reason for choosing the period 1979-2014 beside that it maximizes the trend found in an on longterm average trend less time series?**

The period 1979 – 2014 is when the "high quality passive microwave" sea ice concentration data is available, so our confidence in September SIE is higher for this period than earlier.

**p8, l19: in 2011 and 2012 the spring and winter exports are of similar magnitude but not in 2013 and 2014. Exports were on more similar magnitude during the 1940-50s. The reduction in seasonal cycle therefore is only temporarily.**

Yes, correct. A change will be implemented. We were thinking of the smoothed values here.

**p9, l3: I cannot see that Kwok, 2009 uses reanalysis data. They use satellite data.**

Kwok (2009) used reanalysis data during the summer months, when the passive microwave data does not allow for "proper" feature tracking. Have been changed now.

**p10, l13-14: In Fig 4 the 1995 export is larger than in 2012. That was also correctly stated before.**

The difference comes from the use of calendar year. The Fig. 4 values use 1.September – 31. August.

**p11: see also Kwok et al., 2013 for a detailed discussion of AO and ice circulation.**

We will include both Hillmer and Jung (2000) for the NAO, and the newer Kwok (2013) for the AD/AO comparison. In addition, the work by Wu et al (2006), which come to similar conclusions to ours on the AD link, will be included into the discussion.

**p11: the purpose and conclusions from 4.3 regarding this manuscript remain a bit unclear to me. Better motivate or remove.**

A discussion of the large-scale atmospheric circulation was requested by previous reviewers, and relates to the comparison with the long-term variability simulations by Zhang (2015). A better motivation will be included.

**p13, l8-9: this is a very strong assumption (no feedbacks considered) and makes**

Interactive
comment

**all conclusions based on this more hypothesis and speculations. Not a problem but should be clearly called that then and not presented the same way as the results based on the export time series. Could be more like an outlook section.**

We are here trying to isolate one feedback – the strong ice-albedo feedback. While this is a simple way to do it, we at least clearly state this assumption here, and given that no dedicated simulations are available it is the best thing we can do in our minds.

**p13, l26-31: again speculative; the correlation of -0.43 is modest as you correctly say.**

Yes – correct. We do state that this is a "possible explanation", and that there could be others. This is the discussion section, and we will add more clarification here and speculations should be OK as long as they are clearly stated.

**p14, 4.6: here you estimate the influence of one feedback. But there are many others. See e.g. the influence of ice convergence along the CAA contributing to the 2012 minimum. As a fully coupled system I am not sure one can simply separate feedbacks and sum them up again in the end. All feedbacks will interact with each other, there are many non-linear responses. A coupled GCM would be a better approach to evaluate this.**

We do agree that dedicated regional simulations could potentially be valuable. What we have at hand are the long runs of the GFDL coupled GCM, which largely confirms that the AD is linked to the export, and further that the export is linked to the September SIE. The other factors that are correctly mentioned here were shortly mentioned on Page 15 (line 21-23).

**p15, 4.7: here you look at a GCM but only in relation to AD. Does the GFDL model show high correlations between spring export and summer ice area minima?**

In the 3600 year control simulation of GFDL coupled GCM, the spring Fram Strait ice area export also has an anti-correlation with the September Arctic SIE ($r = -0.34$). This

is similar to the anti-correlation between the observed de-trended spring Fram Strait ice area export and September Arctic SIE (r = -0.43).

This modeling result is also consistent with our conclusion that "the recent increased spring export can directly explain around 10% of the observed September SIE loss". However, a detailed analysis of the GFDL simulations was requested removed by another previous reviewer. We will add some of this text back as it probably has backed up our understanding of these processes, but been removed, so that it seemed more speculative than it really was.

New citations:

van Angelen, J. H., M. R. van den Broeke, and R. Kwok (2011), The Greenland Sea Jet: A mechanism for wind‐driven sea ice export through Fram Strait, Geophys. Res. Lett., 38, L12805, doi:10.1029/2011GL047837.

Walsh, J. E., Fetterer, F., Scott Stewart, J. and Chapman, W. L. (2016), A database for depicting Arctic sea ice variations back to 1850. Geogr Rev. doi:10.1111/j.1931-0846.2016.12195.x

Williams, J. Tremblay, B. and Newton, R. Dynamic preconditioning of the September sea-ice extent minimum, J. (2016), Journal of Climate, 29(16), 5879–5891. DOI: 10.1175/JCLI-D-15-0515.1

Wu, B., Wang, J., Walsh, J. E. (2006). Dipole anomaly in the winter Arctic atmosphere and its association with sea ice motion. Journal of Climate, 19(2), 210-225.

[Figure]

**Fig. 1.** Mid March sea Ice Concentration in the Fram Strait spatially averaged over 15 W to 5 E
using data from Walsh et al (2015).

---

## Author Comment (AC3) · 16 Aug 2016

Thank you for your positive and helpful comment.

We will indeed follow your suggestions, and have started implementing the requests. We certainly agree that "no particular record is the truth" and that the errors "are not well defined", but will do our best to improve the description and discussion as suggested by you and the two reviewers. The answer to your more detailed questions will be given in the author response, as part of the re-submission.

In short we will:

1) Better describe the uncertainty both in the SAR based and "mSLP" based time

series.

2) Extend the existing discussion about the large-scale atmospheric forcing incorporating the suggested Hilmer and Jung (2000) and Wu at al (2006) papers.

3) More carefully include the uncertainties in the discussion of our mSLP based time series before 1979, including long-term changes in sea ice concentration from Walsh et al (2015).

4) Tighten the discussion of the link to the following September SIE, and instead cite the newly published Williams et al (2016) paper.

On behalf of all co-authors,

Lars H. Smedsrud

---

## Author Response (AR1)

Editor Jennifer Hutchings and editorial staff
Copernicus Publications
Editorial Support
editorial@copernicus.org

Bergen, Norway, September 16. 2016

**Author Response to Editor and Reviewers for MS No.: tc-2016-79:**
"Fram Strait sea ice export variability and September Arctic sea ice extent over the last 80 years", by L. H. Smedsrud et al.

Dear Editor. Thank you for your helpful comments, and the chance to resubmit the paper in a substantially changed version. We have indeed followed your suggestions as outlined below, and our response here is given **using bold text**. The new text in the manuscript is shown **using red bold font**, and this version is included at the end of our response letter, on page 13 onwards. We certainly agree that "no particular record is the truth" and that the errors "are not well defined", but have done our best to improve the description and discussion as suggested. In short we have:

1) Better described the uncertainty both in the SAR based and "mSLP" based time series.

2) Extended the discussion about the large-scale atmospheric forcing incorporating the two papers of Hilmer and Jung (2000) and Wu at al (2006).

3) More carefully estimated and discussed uncertainties of our mSLP based time series, including using the long-term changes in sea ice concentration from Walsh et al (2015).

4) Tightened and re-written much of the discussion of the link to the following September SIE, and instead cited the newly published Williams et al (2016) paper.

Lars H. Smedsrud, Mari H. Halvorsen, Julienne C. Stroeve, Rong Zhang and Kjell Kloster.

**Editors Comments:**

Both reviewers express concerns about the presentation of uncertainty in your manuscript. In response to reviewer 2 could you please discuss how the uncertainty in ice drift estimate impacts the area flux estimate. Yes, I agree the variance is not the same as the error estimate, however it is very hard from your current text and figures to gauge the signal to noise ratio. In the context of the trends and whether you can identify periods of variability that are longer than the interannual variability this variance does impact the length of time series you need to draw conclusions. Your manuscript could be clarified in this respect, which would of course make the paper more readable and accessible. For example, you discuss in detail the difference in trends between various different analyses, but do not put this into the context of if these differences are statistically insignificant. You do discuss the point that the last 30 years is impacted by several high export years at the end of the time series, and I think you can do a better job of putting this into context. The fact that your SLP based export estimate differs from previous work needs to be considered in the context of no particular record being the truth and all having errors that are not well defined.

**The uncertainty of the SAR based (2004 – 2014) time series have now been better described in section 2.1. This is the main difference between our new results and previous estimates of the area export. The uncertainties for the mSLP based time series are more difficult to describe, but we have now used the Walsh et al (2015) data to come up with a good estimate in section 4.**

There are several previous works linking the Fram Strait export or the Arctic sea ice pack state and ice motion to the Dipole Anomaly. Please consider the work of Jia Wang's group for example.

**We have added the Wu et al. (2006) paper, which is also a substantial paper on the AD link to export into the discussion in section 4.3. We did previously already cite the Wang et al (2009) paper.**

Your choice of calling March to August Spring is a little unusual in my mind. I understand that you choose this time period based on the assumption all ice that grows in open water created by the export between these times will melt out in summer. This is a highly simplified model and does not account for ridging, but then again the albedo feedback will amplify the melt so you are not really looking at export from March-August as a linear indicator of open water in summer. Did you choose to split the time series periods in March as this is the 6 month split that gives you the best fit of the export to end of summer ice extent?

**The split from March to August was done at the start of the analysis, based on the timing of the maximum extent. It is thus based on simple physics and conveniently splits the year into two halves. Export anomalies from September – February should have a qualitatively different effect. We agree that it is a highly simplified model to estimate the open water created by the export, but this was the basis for the previous section 4.5. Section 4.5 and 4.6 and 4.8 have now all been merged and compacted into one, the new section 4.5.**

I am having trouble wrapping my head around how the over 80% variance in ice export explained by cross strait pressure gradient and assumed seasonal cycle on ocean currents (at a time when the time series is also experiencing larger export), and the 22-55% covariance of the export proxy and following September sea ice extent allow you to make strong statements about causality.

**We agree that our previous discussion was suggesting this link without substantial evidence, but it has now been confirmed for the 1993-2014 period by Williams et al. (2016). What we add additionally in the new version is a discussion of the longer term period between 1935 and 2014.**

While the proxy record is defendable, I am not sure what contribution export has to ice extent based on the correlations. Does the proxy perform as well earlier in the time series, how much of the reduction in correlation is due to decreased covariance of the three station pressure with ice drift? In fact, the shorter time period is influenced by specific high export years, and you show that the last 7 years of the record are where this happens and influences trends. If you were to chose a similar short time series bracketing the years identified by for example Son Ngheim et al. (2007) as high transpolar ice drift and export (e.g. 2005-2007) would you get increased correlation based on this particular event? It does not look so from my quick scan of your figures, and the ice export at Fram Strait lagged (by a year) the transpolar drift event that the 2007 minimum was related to. It appears to me that the only time when the export explained a significant portion of the September ice extent has been in recent years (2011-2014). Is this the case? I agree with reviewer 1 that you should tightening your manuscript to not overstate results.

**We have substantially modified the discussion on the link to the following September SIE, and base some of it on the new Williams et al, (2016) paper now. See the new section 4.5. It is indeed correct that for individual years other factors like ridging probably have a much larger role, and we have included the new Hutchings and Perovich (2015) paper discussing the 2007 event in special. It remains clear that the link appears stronger in recent years, but we do find a modest influence in the long-term, which is physically sound in our view and backed up by the correlations.**

Specific Points

Abstract line 18 and 20: Missing squared from your area dimensions. Also at page 12, lines 20 and 21. Check throughout please.
**We have checked throughout that the sea ice area export values are given as, for ex. 300,000 km².**

Page 2, line 32 "FShas" -> "FS has"
**Corrected now.**

line 25-26 "should be considerably more accurate that 10%". Did you not actually estimate this? I think you just need to reconsider your grammar here.
**The description of the accuracy has been improved, and is now stated in cm/s.**

page 5 line 16-19 You have noted an increased ice drift in winter. Echoing reviewer 1, there is increased open water in summer and potentially changes to surface roughness of the ice. This will impact stress transfer between the wind and ocean, and increased wind stress transfer to the ocean might also lead to increased currents. This is an example of speculative discussion where you could strengthen the manuscript by focusing on your key result (the time series) and a more rounded acknowledgement of its limitations.

page 16 line 25: There is a missing word
**"export" added.**

Please check all your references are listed. I could not find Krumpen et al. (2016) for example.
**All references have been checked and updated now.**

Reviewer 1 Comments:

The manuscript discusses the Fram Strait sea ice area export over the last 80 years, i.e., from 1935 to 2014. Large variability but no longterm trend is found. However, during the last decade according to the presented time series, ice area export increase. The authors, based on comparisons between spring ice export anomalies and summer minima, conclude that the increased ice export is partially responsible for the accelerated decline in Arctic sea ice extent. The variability and long term trends of the Arctic sea ice export and its connection to changes of the sea ice area within the Arctic Basin is an interesting and important topic.

 For the manuscript at hand I had many problems reviewing it because it (a) discusses and mixes very different datasets and methods, and (b) draws very bold and far reaching conclusions based on quite simplified assumptions not taking the complexity of the coupled ocean-sea ice-atmosphere system enough into account:

- the authors construct a Fram Strait sea ice area flux proxy time series based on the across Strait air pressure gradient between Greenland and Svalbard. A regression between a high resolution SAR based ice area flux time series for 2004-2014 and the pressure gradient is performed. The regression coefficients (including a seasonal cycle adjustment) are used to reconstruct the sea ice area flux based on pressure observations alone. No sea ice observations are used before 2004 but only the air pressure. This fact was not initially clear to me as a reader from the methods section and I only understood it from the side note on page 9. Before the authors mention a new longterm sea ice extent time series (Walsh et al., 2015) but in the end they do not use it. This means that the time series before 2014 does not include any variability due to the changing sea ice area within Fram Strait. While Fram Strait is one of the areas in the Arctic with the smaller sea ice decrease during the satellite era it still shows a significant decrease. The time series presented here does not account for any such changes before 2004. These issues or other limits of the proxy time series are not discussed in the manuscript. On the contrary the authors never call it a proxy time series. These facts should be clearly mentioned already at the beginning of the document.

**We have now re-written both the abstract and the methods section, and make it absolutely clear from the start that we rely on the SLP values to construct our 80 year long time series. Previously we tried first to focus on the recent high export years (2004 – 2014), but we agree that it is more clear now.**

**While our method is not standard we have clearly stated what we did in section 2.3. The term "proxy" is usually used for paleo observations like different organisms found in sediment cores that in some way reflect for example surface temperature. The physical relationship between SLP and ice drift is strong and qualitatively very different to this use of the word. We thus used the term "mSLP based" to describe our ice export estimates prior to 2004. This term was used in section 3.1 for example. Note that also "observations" of sea ice cover are some kind of "proxy" in the way that what is really measured by the satellite is radiation, and the uncertainties are in this case also difficult to properly nail down.**

**The fact that we use observed surface pressure is now made clear also in the abstract, so we think this and the other revisions done should make this clear.**

- the Walsh et al. sea ice extent time series covering the complete 1935-2014 period is used for comparisons between ice export and ice extent in the manuscript. For a revised version of the manuscript this dataset should be combined with the air pressure data to add some ice extent variability to the ice export time series, which should make it more realistic. It is unclear to me why this was not done. The Walsh et al. ice extent dataset is prominently introduced as a new and improved time series.

**Yes - we used the new Walsh et al (2015) data set primarily to evaluate effects of sea ice export, because we wanted to investigate September SIE variability in relation to ice export. It is not straight-forward to combine it with the SLP observations to make a new and more 'realistic' ice export because it only provides a mid-month ice concentration field, and many of the winter months have values based on spatial and temporal interpolation. For 2004 – 2014 we use ice concentration for the same days as the SAR imagery. We have now analyzed sea ice extent and ice concentration (Figure 1 and 2), and used that as a basis for new estimates of uncertainty.**

- the 2004 to 2014 part of the time series is based on ice area flux estimates based on manually derived sea ice drift from high resolution SAR imagery. This should give very good estimates of the ice area export. I still would have appreciated some discussion of potential uncertainties due to the manual extraction by a human analyst or how they were mitigated. For example, were the number and the spatial distribution of the manually derived ice drift vectors constant for the complete time series? It is my understanding that this time series was build up over many years. Can we assume that the quality is constant over time? The stated uncertainty of +-3 km for an individual ice drift vector is actual much higher than what I would have expected. The grid cell size of the SAR data is about 100m. Adding some uncertainty caused by geolocation variability and identifying the exact same point in two images I still would have expected an uncertainty on the order of 500m or better like for example reported for the Radarsat RGPS data.

**The SAR time series has images every three days for the 2004 – 2014 period, and have been manually derived by the same person, Kjell Kloster, for all that time. The details are described in a report; Kloster and Sandven (2014). Although it is manually derived, having the same person doing it should lead to a constant quality over time. An independent test of a SAR image pair by the University of Tasmania (Petra Heil, personal comm. 2012) showed that a computer image tracker could re-produce about 60% of the velocity vectors, but gave basically the same vectors for those that were picked up. We have added a better description of the uncertainties in section 2.1 now.**

- the authors then merge the air pressure proxy time series with the SAR based time series. The complete air pressure based ice export time series is not shown. In my opinion that should not be done. The two time series have very different error bars and characteristics. The air pressure gradient is the only information we have got to estimate the ice export before 1979 when the satellite data start. This is argument enough to use the air pressure as a proxy to derive and discuss the ice export variability.

But again, it then also should be clearly stated what kind of time series is discussed in the manuscript. There is quite some focus on the 2004-2014 SAR dataset but the authors state themselves that this time period is too short to discuss significant trends. On page 7 the trends for the 1935-2014 air pressure time series alone are given and it is argued that these statistics are very similar to the merged time series. I would argue the other way around: use a consistent time series, i.e., the air pressure proxy ice export, for the complete period. This will avoid any biases, changes in statistics etc. due to the merging process in 2004.

**We agree that this is an important question, and it is exactly why we discussed this merging in three different paragraphs (Page 7, line 23 – 29, Page 8, line 13 – 33). We did however end up on the opposite conclusion that the best thing was to present the "best possible" merged time series. The trends would be very similar if we should follow this suggestion and plot that in Fig. 2 instead. Analyzing ice concentrations from Walsh et al (2015) for 1935 onwards we find small and not significant trends for most months. The two figures below shows this clearly.**

[Figure]

[Figure]

**Figure 1: Sea ice concentrations in the Fram Strait from the Walsh et al. (2015) data. Spatial average between 15W and 5 E. a) Mid-month sea ice concentration for January between 1935 and 2013. b) Seasonal means for 1935 – 2013. The spring trend is – 0.1 %/decade, and the winter trend is -0.9 %/decade, but they are not significantly different from zero (p>0.05).**

Figure 2 shows the similarity of the seasonal cycle between the adapted air pressure and SAR ice export time series. This is nice and shows good agreement but also differences for some months. For the reader it would be important to also see the two time series together for the complete 2004-2014 period. If the complete discussion in the manuscript would be changed to the air pressure only time series (see my last point) the SAR derived time series could be added to Fig. 4 for comparison.

**We understand the importance of checking the agreement between the mSLP based and SAR based values. Fig. 4 in Smedsrud (2011) shows such a comparison for 2004 – 2010. The updated values are similar, and we found no particular reason to include them as a separate figure here. From visual inspection of Fig. 4 here it should be clear that there are no significant differences in the merged values on either side of 2004.**

- The manuscript mentions that their ice export estimates for the last 30 years do not agree with estimates from passive microwave radiometers (e.g. Kwok et al., 2013). Actually, these satellite data based time series do not find a trend in ice export, which is opposite to the trend found here from the air pressure data. The authors attribute this difference to the low resolution of the satellite data and that it will not correctly track all ice in Fram Strait (p. 12). That is one possible explanation but the authors do not demonstrate this failure but hypothesis it. That is okay because the satellite data is not the topic of their study. But then the authors should be more critical also towards their own time series and list factors, which could explain the difference to the satellite data. For example: there is an increase in the across pressure gradient during the last 30 years. As this is the only data used in the proxy ice export time series presented here this directly results in a positive ice export trend.

However, there are other factors, which influence Fram Strait ice export and could or have changed during the last decades and therefore counteract the increased pressure gradient:

(i) the ice area in Fram Strait (FS) shows a negative trend reducing the ice area export, which is not accounted for here.

(ii) the surface winds in FS are not only determined by the pressure gradient but have a strong contribution from thermal wind (THW) forcing (van Angelen et al., 2011). If the THW forcing would have been reduced during the last decades that would counteract the increased pressure gradient

(iii) the ice surface drag (surface roughness) could have changed, i.e., the atmosphere to ice energy transfer function can have changed. This could also be caused by a change of internal ice stress, i.e., how lose or compact the ice in FS is.

(iv) the ocean forcing can have changed

I don't know if these factors can explain the difference to the satellite ice export time series but they should be discussed. Also in the summary it should be mentioned that all conclusion drawn here are based on the air pressure time series presented but that for other available ice export estimates one would get to complete opposite conclusions.

**The reviewer states an important point, and we have not tried to "minimize" the sea ice export variability not related to SLP. We agree that there are a number of physical parameters that could have changed over these 80 years, and we have added a better discussion of these points in the new version. All four points are valid, and i) have been quantified based on the Walsh et al (2015) data in Fig. 1 above. We have further extended the discussion of uncertainties based on the Walsh et al. (2015) data below in figure 2. While there is no way to acquire more detailed observations, the Walsh et al (2015) data does provide a simple guide to uncertainties in some of these parameters. The 10-20% variability in March ice concentration translates into a variability of about 10% in sea ice export.**

**We also agree that the stronger thermal wind forcing during winter (van Angelen et al. 2011) is another explanation for the larger export during winter than estimated by the mSLP. We previously discussed this seasonal difference and attributed it to a**

**stronger East Greenland Current (EGC, page 5 line 10 – 28). It is also consistent with a stronger thermal wind, and this has now been added. Note that the simulations of van Angelen et al. (2011) did not include an ocean model, so the thermal wind could well explain the stronger current during winter.**

[Figure]

**Figure 2: The Fram Strait Ice Area Export in March. The red line shows the SAR + mSLP based time series, while the black line shows the effect of variations in March sea ice concentration, using the mean March Export of 113,000 km².**

**For iii) the main influence on roughness is probably sea ice concentration as discussed above for the Walsh et al. (2015) data of, and changes thickness. It is likely that before 2004 ice was thicker and moved less effectively for a given mSLP gradient as found by Kwok et al (2013). This would lead to smaller values of ice export prior to 2004, and would thus increase the trend onwards from 1979. Our results remain different from for example Fig. 7 of Kwok et al (2013) that finds a positive trend for summer ice area export (June – September), but not for the annual values. Likewise is a change in ocean forcing iv) possible, but not observations are available to discuss such a change.**

**The main difference from Kwok et al (2013) is the 2004 – 2014 time period when we have higher export values. In this period we use the observed passive microwave sea ice concentration. This is in short why we wanted to present the "best possible" time series and not the "mSLP based" time series as suggested above.**

- In section 4 from 4.2 onwards the sea ice area export time series and the Walsh et al. sea ice extent time series are used to draw quite far reaching conclusion about the influence of the sea ice export increase they find on the recent decrease in Arctic sea ice area. They make the in my view oversimplified assumption that every spring ice export anomaly directly relates to a loss in ice area for the summer sea ice extent. There are many other factors which will influence this relationship, e.g., if the ice gets compressed or more spread out in the Arctic Basin and many more feedbacks the authors are well aware of. One would need a

coupled Arctic regional climate model to make more robust conclusions about such relationship. I actually like such simplified speculations in the way of: "If we would assume the ice export anomalies to directly relate to anomalies in Arctic summer ice area this would mean . . ." But here they are presented as hard results and in a very broad way. I recommend to remove most of the discussion related to this in section 4 and concentrate on the new 80 year ice export time series at hand. Some of these hypothetical consequences can then be briefly mentioned at the end of the discussion.

**We understand the reviewers point. Specified simulations using a regional climate model could be performed for another way of estimating the effects of the sea ice export variability. Such model simulations are complicated, and have not been performed. Using a dynamical sea ice drift model Williams et al (2016) have actually performed experiments using coastal divergence and Fram Strait export, and find a similar level of influence on the September SIE. We are indeed aware of many other factors influencing September SIE variability, and only stated that between 18% - 22% is caused by the export, apart from in the last 10 years. Our understanding is also based on the long control run from the coupled GFDL model. In a previous version of this paper (Halvorsen at al 2015, The Cryospere Discussions) these model results were included in more detail, and backed up our understanding. They were subsequently removed due to a previous reviewer's suggestion.**

**We have rewritten the discussion, removed the speculative parts, and added more text about the link in simulations between September SIE and FS ice export in section 4.6.**

The 80 year long air pressure based FS ice export time series by itself merits publication. Some information about the actual sea ice variability from the Walsh et al. dataset should be added. Errors and uncertainties have to be discussed more upfront and also in relation to other published but much shorter ice area export estimates. The mainly speculative discussion about consequences should be reduced and declared more clearly.

**Thank you for your interest in the export itself. We agree, but also found that more people are interested in the export if the plausible consequences are also discussed. This is what we attempted to do here.**

Minor comments:

p7, l18: for 2011-2013 the export exceeds 1mil sq km.
**Corrected.**

p8, 3.2: is there a reason for choosing the period 1979-2014 beside that it maximizes the trend found in an on longterm average trend less time series?

**The period 1979 – 2014 is when the "high quality passive microwave" sea ice concentration data is available, so our confidence in September SIE is higher for this period than earlier.**

p8, l19: in 2011 and 2012 the spring and winter exports are of similar magnitude but not in 2013 and 2014. Exports were on more similar magnitude during the 1940-50s. The reduction in seasonal cycle therefore is only temporarily.
**Yes, corrected. We were thinking of the smoothed values here.**

p9, l3: I cannot see that Kwok, 2009 uses reanalysis data. They use satellite data.
**Kwok (2009) used reanalysis data during the summer months, when the passive microwave data does not allow for "proper" feature tracking. We now cite the new van Angelen et al (2011) instead, that use re-analysis as boundary conditions.**

p10, l13-14: In Fig 4 the 1995 export is larger than in 2012. That was also correctly stated before.
**The difference comes from the use of calendar year. The Fig. 4 values use 1.September – 31. August.**

p11: see also Kwok et al., 2013 for a detailed discussion of AO and ice circulation.
**We have included both Hillmer and Jung (2000) for the NAO. In addition, the work by Wu et al (2006), which come to similar conclusions to ours on the AD link, has been included in the discussion in section 4.3.**

p11: the purpose and conclusions from 4.3 regarding this manuscript remain a bit unclear to me. Better motivate or remove.

**A discussion of the large-scale atmospheric circulation was requested by previous reviewers and reviewer 2. This also relates to the comparison with the long-term variability simulations by Zhang (2015).**

p13, l8-9: this is a very strong assumption (no feedbacks considered) and makes all conclusions based on this more hypothesis and speculations. Not a problem but should be clearly called that then and not presented the same way as the results based on the export time series. Could be more like an outlook section.

p13, l26-31: again speculative; the correlation of -0.43 is modest as you correctly say.

**Section 4.5 has been substantially revised. The new section 4.5 is a much condensed version of the previous section 4.5, 4.6 and 4.8.**

p14, 4.6: here you estimate the influence of one feedback. But there are many others. See e.g. the influence of ice convergence along the CAA contributing to the 2012 minimum. As a fully coupled system I am not sure one can simply separate feedbacks and sum them up again in the end. All feedbacks will interact with each other, there are many non-linear responses. A coupled GCM would be a better approach to evaluate this.

**We do agree that dedicated regional simulations could potentially be valuable. What we have at hand are the long runs of the GFDL coupled GCM, which largely confirms that the AD is linked to the export, and further that the export is linked to the September SIE. The other factors that are correctly mentioned here were shortly mentioned on Page 15 (line 21-23). The old section 4.6 was condensed into just one sentence in the new section 4.5.**

p15, 4.7: here you look at a GCM but only in relation to AD. Does the GFDL model show high correlations between spring export and summer ice area minima?

**A detailed analysis of the GFDL simulations was requested removed by another previous reviewer. Some of this text has been added back now in the new section 4.6 and this perhaps explains partly why this discussion seemed more speculative than it really was.**

Reviewer 2 Comments:

This paper attempts to extend the time series of Fram Strait (FS) ice export back to 1935. My primary concern with this paper is the accumulated errors in their regression of ice velocities going back to 1935. Given these uncertainties, I don't think they can make any definitive conclusions based on the extrapolated time series. Details on this concern and other comments are provided below. I suggest rejection of this paper.

1) Standard error about the regression line for equation 1.
The authors state a standard error of the regression line of 3.4 cm/s. Ice velocities are typically 12 cm/s. The error adds up to an ice export uncertainty of +/- 250000 km^2, which is also the variance about the mean of 883000 km^2. Given this uncertainty, it is hard to trust any conclusions drawn on their extrapolated time series, which is foundation of this paper.

**The standard error is a statistical estimate of uncertainty, and describes the scatter around the regression line. The scatter is caused by the other factors influencing sea ice drift other than the geostrophic wind, and will be close to normally distributed around the regression line. The method is the same as used in Smedsrud et al (2011), but with 5 years of extended data. The uncertainty will further decrease when averaging into seasons is performed, because some months have slightly higher speed than the regression predicts, and some will have lower. So it is not correct to "add up" the uncertainty as suggested by the reviewer here.**

**The new study be van Angelen et al. 2011 also find that the SLP gradient is a very useful estimate of Fram Strait Export. Uncertainties for 1935 - 2004 have now been estimated using the Walsh et al (2015) data on ice concentration.**

2) Fram Strait SLP Gradient
I think the linear interpolation to estimate pressure to 78N after 1958 is probably OK since the stations are close, but prior to 1958, the southern station may be too far?
The authors need to substantiate the use of the 3 weather stations on Greenland to interpolate SLP at 78N and estimate the across strait pressure gradient. One way to do this is to compare the estimated SLP at 78N based on the regression from Nord to Danmarkshavn, and Nord to Tasillaq during a period when they have data from all 3 stations.

Equation 1 should also be evaluated based on the 2 estimates of dp/dx across the strait to see how much difference the use of the different stations make.

**We did perform correlations between the stations as described on page 4, line 11-18. There is relative lower but significant correlation (r=0.77 instead of r =0.93) between Nord and Tasiilaq, and there are no other alternative observations available prior to 1958, so this is the best data we have. The SLP pattern tends to follow the Greenland coast quite well (Fig. 4c in Hilmer & Jung (2000) for example), and what matters here is the East-west SLP gradient, which should be robust. No particular change in variability is visible in Figure 4 before and after 1958.**

3) The authors should cite Hilmer and Jung, 2000 "Evidence for a recent change in the link between the North Atlantic Oscillation and Arctic sea ice export", in any discussion of Fram Strait ice flux. I think this is the definitive paper on the topic. Given that HJ cover the period going back to 1958, and many of the authors own papers discuss the period after this to the present. This paper would really have to substantiate their estimates for export prior to 1958 to make an acceptable contribution to the literature.

**Thank you for pointing out this paper, it has been cited in the updated version. We find that some of our conclusions are consistent with their results. Prior to 1978 Hilmer**

and Jung (2000) used simulations with quite a coarse resolution numerical model driven by another set of simulations (NCEP reanalysis) that are now known to have several issues in the Arctic. We therefore have more confidence in our own results for the early time period, as they are directly based on observations. Note that both the "missing" link between NAO and winter export, and a (not discussed) long term trend 1958 – 1997 is qualitatively consistent with our results. We used the AO index in our discussion (page 11) instead of the NAO, as it is a better index for the Arctic large scale atmospheric circulation and is also highly correlated with the NAO.

[revised manuscript text omitted]

---

## Author Response (AR2)

Editor Jennifer Hutchings and editorial staff

Copernicus Publications

Editorial Support

editorial@copernicus.org

Longyearbyen, Svalbard, December 6. 2016

**Author Response to Editor and one Reviewer for MS No.: tc-2016-79:** "Fram Strait sea ice export variability and September Arctic sea ice extent over the last 80 years"

Dear Editor.

Thank you for your helpful comments. We are glad to state that we have fully implemented the suggested new data analysis and changes. In short we produced a new figure showing the comparison between the SAR based and mSLP based values for 2004 – 2014 as requested (new Figure 3).  We have also included the effect of the varying ice concentration from Walsh et al. (2015) in an existing that now is Figure 5. We do agree that this improved the clearness and basis for our conclusion in a significant way.

We have also more precisely described the methods so it should be very clear that we have indeed used varying sea ice concentration for the 3-day time periods when SAR images are available, so the "bias" issue raised by the reviewer does not apply. We have also modified the abstract and discussion to reflect the suggested changes.

We have received a number of requests for the ice area export time series, and have added the data to PAGEA now. We will publish this with the new doi from the Cryopshere paper if this new version of the paper can be accepted now: https://doi.pangaea.de/10.1594/PANGAEA.868944

Our more detailed response is given **using bold text** below. The new improved text in the manuscript is shown **using red bold font**.

Lars H. Smedsrud, Mari H. Halvorsen, Julienne C. Stroeve, Rong Zhang and Kjell Kloster.

**Review of "Fram Strait sea ice export variability and September Arctic sea ice extent over the last 80 years" by Lars H. Smedsrud, Mari H. Halvorsen, Julienne C. Stroeve, Rong Zhang, and Kjell Kloster**

This is my second review of the manuscript. I therefore will concentrate on, if my initially major concerns were addressed appropriately. I thank the authors for providing a revised version of the manuscript. The text was changed at many occasions and many specific comments were addressed adequately. The analysis, however, was not changed or extended significantly.

My major concerns for the first version of the manuscript were

1) Before 2004 the authors only use the sea level pressure difference across Fram Strait (mSLP) as a proxy for the sea ice area export and it was suggested to combine the mSLP derived ice drift with ice area data from Walsh et al., which is used for other purposes in the analysis.

- The authors did not follow the suggestion to call the mSLP time series a proxy time series (neither ice drift nor ice area are observed directly here) but now clearly state that it is only mSLP based and clarified that several times in the manuscript, which is fine. They also clarified the description in the data section (which now raised new questions, see below).

**Answer: We are glad the reviewer finds our solution here satisfying.**

They, however, did not combine their mSLP time series with the ice area time series. This would add variability in sea ice area to the the so called "sea ice area export time series", which actually before 2004 only is a (indirect) ice drift speed time series. Ice area information is not included. How variable the ice area in Fram Strait can be is, e.g., shown in Kwok, 2009 (including a negative trend counteracting the positive trend found here).

The author's argument for not using the ice area data is: "It is not straight-forward to combine it with the SLP observations to make a new and more 'realistic' ice export because it only provides a mid-month ice concentration field, and many of the winter months have values based on spatial and temporal interpolation." I cannot follow that argument. In sections 2.1 and 2.2 it is described that (a) even for the SAR drift data monthly ice concentrations are used and (b) the mSLP time series is a monthly dataset. The Walsh et al. mid-month is the best information we have and is used elsewhere in the manuscript. Actually, Fram Strait is one of the better observed areas in the Arctic for the pre-satellite area because of ship logs.

**Answer: We have now implemented the effect of changing sea ice concentration over time. The effect is added as an additional time series for the winter and spring seasonal ice export in Figure 5.**

**As described further below are we using 3-day mean values for both sea ice concentration and ice drift for the 2004 – 2014 time series. The later averaging into monthly means lowers the uncertainty of these time series, while this would be fundamentally different for the Walsh (2015) data that are only available as mid-month values. We therefore used seasonal mean anomalies for sea ice concentration, and estimated the corresponding effect on the ice export. This shows that there is an effect in the 1950's, but that the differences are small after 1979. Because the mSLP values are based daily observations we still hold the original time series as "the best", but we are glad that the time-varying effect of sea ice concentration could now be included as well.**

2) Uncertainty of SAR derived ice drift time series.

- More information was added to the manuscript and some estimates of expected uncertainty presented. Thanks. (I had a problem to find the Kloster and Sandven (2014) reference, see below)

**Answer: Thanks for pointing out the unclear text. The last few details confirming that we do use 3-daily values of sea ice concentration is now added. The url for the available report has been added to the reference list.**

3) Two very different time series are merged, the mSLP proxy based and the higher resolution SAR based, which also includes ice area variability. The complete mSLP based time series should be presented in addition to the merged one including statistics to let the reader judge by themselves about the differences.

- The authors stick to presenting "the best possible merged time series" only. Instead they present in their reply (for reasons not completely clear to me) the Walsh et al. 1935-2013 sea ice concentration time series, which actually just proves my last point that there is quite some variability in sea ice area in Fram Strait and including it in the ice area flux time series should be beneficial.

For the mSLP to SAR time series they refer to Smedsrud et al. (2011). Fig. 4 there shows the difference between an NCEP SLP based time series and the SAR one but not the observed SLP based time series used here. I would have loved to have such a comparison plot also in this manuscript. Fig. 4 there actually shows, in my opinion, quite high deviations between the two datasets. In summer deviations can exceed 50% in certain years and in winter deviations still can be at the 25% level. Fig. 2 in this manuscript only shows the mean seasonal cycle including standard deviations, which does not allow any conclusions on the differences, i.e., error of the mSLP based time series if we assume the SAR based one to be the truth.

**Answer: Thank you for this constructive suggestion. We have now included a new Figure 3 showing both the mSLP based time series together with the "best possible" SAR based time series for 2004 – 2014. The differences are smaller than in Smedsrud et al (2011) because we corrected for the seasonal differences due to the ocean current, or perhaps the atmospheric jet, as discussed.**

4) Ice area fluxes presented here are significantly different from previous estimates based on satellite radiometer data. Conclusions drawn from these time series would be opposite than what is presented here. The quality of the time series presented here was not quantitatively assessed against previous estimates. Conclusions therefore carefully be drawn based on the presented time series and other possibilities and explanation be mentioned.

- The authors give an extensive answer on this concern, which is appreciated. However, not much of it made it to the manuscript. The van Angelen paper is now mentioned and one short paragraph (p5, l38-41) was added at the end of section 2. The additions actually do not discuss differences to previous area flux estimates and no critical discussion was added to the results or conclusion section. While the authors in the first paragraph of their reply "certainly agree that "no particular record is the truth" and that the errors "are not well defined""

The plots in Fig.1 of the reply are actually interesting. I agree that there is no significant trend in ice area for 1935-2013 but there seems to be a clear negative trend after 1955 and also for the 1979-2013 period discussed in more detail here. In the reply Fig. 2 the authors present the mSLP based time series together with some ice

concentration variability. The conclusions from this are not completely clear to me besides that the two time series do not co-vary. The only analysis that made it in the new manuscript as far as I can see is that the ice concentration variability only makes up 10% of the ice area flux variability, which is true and was shown before. However, that does not mean that changes in sea ice area cannot offset or reduce a 6% trend in ice area flux.

**Answer: This was a good point to rise, and the effect of seasonal variability in sea ice concentration for Fram Strait (79 N, 15W – 5E) have now been analyzed and added to Figure 5. The effect is less than 10%, as suggested. The adjusted values would reduce trends between the 1950's and the 1980's. However, trends after 1979 remain unaffected.**

5) Very far reaching conclusions about the fate of the Arctic ice cover are drawn based on a very uncertain time series and very strong simplifications of physical processes and their interaction in the coupled climate system.

- The authors shortened the discussion section, added more references but rebutted that the simplifications might lead to overstated consequences. I still think the conclusions are too general and not clearly enough marked as hypothetical because they are (a) based on a uncertain time series with contradicting results found by other studies and (b) left out many physical processes, e.g., ice deformation and many others. I would prefer an approach where first the quality of a new time series is assed rigorously and discussed critically before far reaching conclusions are drawn or the limitation are clearly mentioned in the discussion. Anyway, this is probably also a matter of personal taste and one can agree to disagree.

The text of the manuscript has improved. I also appreciate the shortening of discussion section 4.5 and 4.6 while I still don't agree on the, in my view, over-simplifications in assumptions applied here. However, most of my main concerns (see above) were not addressed. In addition, the improved new method section raised another question about the method and a potential bias. I do not have the feeling the authors included mayor revisions to address the two reviewers and editors concerns. It is the right and has my full understanding that authors rebut reviewers comments they don't agree on. But I hope I could clarify why I cannot agree on certain points. I therefore stand with my initial assessment that the manuscript needs mayor revisions before publication.

**Answer: Thanks for confirming that the manuscript has improved. We have now addressed the two major concerns as explained above. We agree that this made a major improvement to the manuscript in the way that these issues of doubt could now largely be removed.**

Additional major comments

p3, l15-16: If I understand the procedure in the last paragraph correctly you first calculate a monthly mean sea ice drift speed value for the complete Fram Strait, i.e., you have one ice speed value per month. In a second step you now multiply this one ice drift speed value with one monthly mean sea ice concentration value for the complete strait.

**Answer: We use 3-day means of ice concentration. So the problem suggested below does not apply.**

Your figure 3 shows that drift speeds are much higher in the east than in the west of Fram Strait. Ice concentrations are, however, are much lower in the east. You have to multiply your high drift speeds in the east with low ice concentrations and your low drift speeds in the west with high ice concentrations to get the correct

ice area flux. Your method will yield a positive ice area flux bias, which might explain some of your higher flux estimates compared to previous studies. If this bias is constant in time is not known and therefore can, potentially, also influence the area flux trend. This issue should be investigated.

In your replies to my comments you write on p5 that you use daily ice concentration data in contrast to what is written here. Such contradictions leave open questions to me about the method applied.

**Answer: We now specify in more detail that we do use 3-day ice concentration values in section 2.1 for the SAR based time series. The trends and export since 1979 also showed to not be sensitive to changes in sea ice concentration from Walsh et al. (2015) as shown in Figure 5.**

**Minor comments**

p2, l15-20: Another recent study does not find a significant Fram Strait Sea ice area flux trends for 1988-2012 (Positive trend in summer, negative in winter).

Bi, H., K. Sun, X. Zhou, H. Huang & X. Xu (2016): Arctic Sea Ice Area Export Through the Fram Strait Estimated From Satellite-Based Data: 1988–2012, IEEE Journal of Selected Topics in Applied Earth Observations and Remote Sensing, 9, 3144-3157, doi:10.1109/jstars.2016.2584539.

**Answer: This new paper compares existing NSIDC data with the Smedsrud et al (2011) values, and finds very similar conclusions for the atmospheric drivers of the ice export. They do not find a trend for the shorter 1988 – 2012, but uses different "winter" and "summer" means, and cannot explain the differences between SAR based and passive microwave based values. They do, however, confirm that the SAR based values should be better. We thus find no important reason to include a citation this late in the review process. We will make our data available to the authors, and perhaps cooperate in the future to explain the differences better.**

Fig. 3: caption says dataset until December 2013, text says until December 2014

**This has now been corrected.**

**References**

I could not verify any statements regarding the SAR sea ice drift dataset (e.g. sec 2.1) because I could not find how to access the main reference:

Kloster, K. and Sandven, S.: Ice Motion and Ice Area Flux in Fram Strait at 79N using SAR and passive microwave for Aug. 2004–Jul. 2014., Technical Report no. 322c. Nansen Environmental and Remote Sensing Center, Bergen, Norway, 2014.

If this technical report is not accessible to the general public it should be removed from the reference list.

**Answer: A 2011 version of the report has been available on the institute web pages, but we have now also made the updated 2015 version available at the url below. This was also added to the citations list:**

**https://www.nersc.no/sites/www.nersc.no/files/NERSC-TecRep-322d(2015).pdf**

---

## Author Response (AR3)

Editor Jennifer Hutchings and editorial staff

Copernicus Publications

Editorial Support

editorial@copernicus.org

Bergen, Norway, December 16. 2016

**Author Response to Editor for MS No.: tc-2016-79:** "Fram Strait sea ice export variability and September Arctic sea ice extent over the last 80 years"

Dear Editor.

Thank you for your helpful comments. We agree that the main focus of the paper is the export values themselves, and that they alone could be basis for the manuscript. However, we also think that when we also include a discussion of the effects of ice export anomalies on September Sea Ice Extent a broader set of readers of the Cryosphere will be interested.

In the interest of time we return this as quickly as possible, and have followed your suggested changes based on the previous review. A more general improvement in our discussion relating to the "missing mechanisms and processes" has not been performed, but some new text has been added mentioning these in the conclusion. In our view we have based our existing discussion well on previously published results, and have stated the assumptions and limitations clearly.

Our more detailed response is given **using bold text** below. The changes in the manuscript are shown using "Track Changes".

Lars H. Smedsrud, Mari H. Halvorsen, Julienne C. Stroeve, Rong Zhang and Kjell Kloster.

**Dear Lars and co-authors.**

Thank you for your response to the review and especially for paying attention to the concern regarding the role of varying ice area in interannual variability in Fram Strait export.

The reviewer did express concern about your results being overstated in the discussion and there not being a clear delineation between the direct findings and hypothesizes about how changes in export drive internal Arctic ice pack extent changes in summer. I also find this, and agree there are missing mechanism and processes in your discussion. I agree with the reviewers that it would be best to focus on the data set presented (your proxy time series) and clearly delineate hypothesized implications of your findings. Below are some specific comments where I find you can draw this distinction.

**Answer: We do follow your argument, and have performed the suggested changes. Our previous text was supported by earlier simulations, largely consistent with the simulations of Zhang (2015), and the new results from Williams et al (2016). If a more substantial alteration of our discussion is required we need more time, but can do this in January 2017.**

In the abstract your newly added text discusses the relationship between export and opening in the central Arctic, and hence summer ice extent. It would be important here to identify that you only find 18% correlation between export and end of summer ice extent.

**Answer: This has now been added to the abstract.**

Section 4.6 is where your discussion delves into territory outside of the scope of your time series analysis. The paragraph at line 30, page 12 is in particular overstating in how you relate your findings to climate model simulations. These models do not show the recent decline in ice extent, and if they do it may not be for the same mechanisms as we are observing now. There is a wide range of literature on this that I am sure you and your co-authors are aware of. No evidence of a trend in export in models between 1979-2013 should not be compared to your result that there is no trend over a longer time period. And no trend does not imply that the phenomena are purely driven by natural variability. It is possible for interannual and decadal scale variability to increase, due to external forcing, while maintaining no trend. In my opinion this section should be reduced to meet the reviewer's concerns.

**Answer: This section was initially included after a request from an earlier reviewer that would like such a discussion and comparison to future global coupled model simulations. We have now reduced the text as requested, and removed the text about the trends and natural variability. We are indeed aware of the wide range of literature, and agree that it is best to remain focused on the export here. We are also quite fine with deleting this last paragraph of 4.6 entirely. This would leave out much of the discussion about the difference between global warming in contrast to "natural variability". What remains clear is that natural variability is significant, and this is what is also stated in the conclusion.**

You have presented a very simplistic model for how increased export relates to summer ice extent reductions. In general I am a fan of simple mechanistic models that can be tested, provided the limitations of the model are clearly articulated. I would caution you to make sure your text is clear that you are describing the mechanism disregarding some potentially important factors (changing stability in the ocean/atmosphere for example), in the interest of presenting the existence of the link and recognize that quantifying the physical mechanisms involved in the link is not possible.

**Answer: We understand your concern. Adding the GFDL model simulations was indeed an attempt at quantifying this link further, and it did confirm this link, at least in part. We have added some text in the conclusion that emphasize clearly other processes leading to September SIE variability, and that this is actually 82% of the variance. We have also modified the text about the increased correlations for the last decades. Correlations do of course not imply any causal relationships, but our confidence is carried by the physical mechanism itself. It is further supported by earlier simulations (Smedsrud et al 2008, Langehaug et al. 2013, and Zhang 2015), and the rather complicated drift model of Williams et al (2016).**

Specific Edits Required

p1,l34: remove duplicate 'the last'

**Answer: Done.**

p2,p13-22: I disagree about not referencing recent papers you were unaware of when writing your manuscript, if they are relevant to your results. Kloster and Sandven (2015) is obviously a relevant study and does enter into the conversation about the large differences in estimates of trends in Fram Strait export.
On this topic, did you consider the differences in location of flux gates in explaining differences between studies?

**Answer: The new Kloster and Sandven (2015) report is the last version, but we have been citing the previous version all the time. Kjell Kloster is maintaining this tracking manually and has normally written a report presenting new values each year. As a co-author he has kept us updated on the comparison of the two flux gates since 2012, and the differences are smaller than 10% (Kloster and Sandven 2015). It is across 79N that we have the longest time series, so this is what we have used here to get to the long-term variability. Because we have been aware of this work all along, and the new report is cited instead of the older version, we have not applied any changes here.**

p4,l26: I realize that you can not identify if there have been changes in seasonality of the fram strait ice drift and export prior to 1979. However if there were changes this will affect your bias correction. It is fair to point out the limitations of your assumptions. p5,l21 might be a relevant place to point this out, or where you introduce the method.

**Answer: Yes – this is a fair point to mention, so we have added a note where suggested (p5, line 21).**

p5,l26: How do you blend the two products? The use of this word makes me think you may be doing more than a simple substition. Please be more precise in your description here.

**Answer: We have not done anything fancy here, we do indeed only add the two time series together. Before 2004 the values are based on the mSLP values only, and afterwards on the SAR. We have changed "blend" to "merge" now, including for the section title. The word "blend" is now only used for the Walsh et al (2015) data that do apply a more sophisticated blending of products.**

I am happy that you have discussed the relative impact on results for both the full proxy time series and the blended time series. I do note that you have chosen to present in figures the time series that demonstrates the largest change in the last decade.

**Answer: Thank you, we agree that this was an important test. We have not used the SAR based values because they demonstrate the largest trend, we used them because they should be the best possible estimate of the ice export since 2004. When calculating trends towards 2014 it would in our view be less correct to use the mSLP based values, but before 2004 they are the best possible values.**

p6,l11: "true monthly mSLP". I am not sure what you mean by true here. I would chose a descriptor that clarifies this is the observed pressure. (truth is not a given for weather station data too, any measurement has an error associated with it!)

**Answer: Yes, by "true" here we meant a proper monthly mean, i.e. a large number of observations as compared to the Walsh et al. (2015) data that are only one mid-month extent value. Changed now, using station-based observations instead.**